# Circular RNA circGlis3 protects against islet β-cell dysfunction and apoptosis in obesity

Yue Liu[1], Yue Yang[1], Chenying Xu[1], Jianxing Liu[1], Jiale Chen[1], Guoqing Li[2], Bin Huang[1], Yi Pan[1], Yanfeng Zhang[1], Qiong Wei[2], Stephen J. Pandol[3], Fangfang Zhang[1] ✉, Ling Li[2] ✉ & Liang Jin [1] ✉

Pancreatic β-cell compensation is a major mechanism in delaying T2DM progression. Here we report the abnormal high expression of circGlis3 in islets of male mice with obesity and serum of people with obesity. Increasing circGlis3 is regulated by Quaking (QKI)-mediated splicing circularization. circGlis3 overexpression enhances insulin secretion and inhibits obesity-induced apoptosis in vitro and in vivo. Mechanistically, circGlis3 promotes insulin secretion by up-regulating *NeuroD1* and *Creb1* via sponging miR-124-3p and decreases apoptosis via interacting with the pro-apoptotic factor SCOTIN. The RNA binding protein FUS recruits circGlis3 and collectively assemble abnormal stable cytoplasmic stress granules (SG) in response to cellular stress. These findings highlight a physiological role for circRNAs in β-cell compensation and indicate that modulation of circGlis3 expression may represent a potential strategy to prevent β-cell dysfunction and apoptosis after obesity.

Diabetes is a rapidly increasing disease worldwide and a substantial threat to human health[1]. Risk factors, including obesity, ageing, a diet containing more refined food and fats, sedentary lifestyles, and genetic factors, all contribute to the accelerating diabetes epidemic[2]. Type 2 diabetes mellitus (T2DM) is the main form of diabetes, especially in China, and T2DM accounts for more than 90% of all diabetes cases. Moreover, some clinical studies have further divided T2DM into 5 subgroups, and various genetic factors have a major role in the pathogenesis of T2DM. Some variants of genes such as TCF7L2, *TM6SF2,* and *HLA locus*, which were previously associated with T2DM, are also linked with the novel subgroups of T2DM[3]. T2DM progresses from compensated insulin resistance to β-cell failure, culminating in uncompensated hyperglycemia[4]. Obesity is frequently a major risk factor for T2DM, but not all patients with obesity develop T2DM. Many patients with obesity do not develop T2DM but adapt to chronic insulin resistance by increasing β-cell mass and insulin secretion. Patients with obesity and T2DM, however, have a ~60% deficit in β-cell mass, and the underlying mechanism is increased β-cell apoptosis[5]. Therapeutic approaches designed to arrest apoptosis might be an important new development in the management of T2DM, since they could reverse the disease rather than just relieve glycaemia. The mechanism underlying the prevention of β-cell apoptosis in the compensatory stage remains unclear.

The adaptation of β-cell mass and function during obesity is associated with alterations in the expression of protein-coding and non-coding transcripts[6]. We previously showed that different types of non-coding RNAs, including microRNAs and long non-coding RNAs, are key players in the regulation of β-cell functions in compensation[7,8]. However, the role of the newly discovered class of circRNAs remains unknown. Based on transcriptomic profiling studies of islets, thousands of islet-specific circRNAs have been identified in human pancreatic islets, most of which were also conserved in mouse islets[9]. *ciRS-7/CDR1as* was the first circRNA studied in pancreatic islet cells. Overexpression of *ciRS-7* in murine islet cells was shown to increase insulin content and secretion[10]. In another investigation, Lisa Stol et al. discovered that *circHIPK3* expression was directly related to insulin exocytosis via coexpression with genes critical for normal islet function, such as *Akt1*, *Slc2a2*, and *Mtpn*[9]. Other circRNAs dysregulated in the

[1]State Key Laboratory of Natural Medicines, Jiangsu Key Laboratory of Druggability of Biopharmaceuticals, School of life Science and Technology, China Pharmaceutical University, 24 Tongjiaxiang, Nanjing, Jiangsu province, P. R. China. [2]Department of Endocrinology, Zhongda Hospital, School of Medicine, Southeast University, No. 87 Dingjiaqiao, Nanjing, Jiangsu 210009, China. [3]Departments of Medicine and Biomedical Sciences, Cedars-Sinai Medical Center, Los Angeles, CA, USA. ✉e-mail: 1620194592@cpu.edu.cn; li-ling76@hotmail.com; ljstemcell@cpu.edu.cn

islets of diabetic *Lepr*[db/db] mice were recently identified using high-throughput RNA sequencing[11]. Although these studies provide the essential groundwork for understanding the function of circRNAs in β-cells, it remains unclear how these circRNAs contribute to obesity-mediated β-cell dysfunction and apoptosis.

We discovered that circGlis3, a circRNA enriched in islets, originated from an exonic sequence of *Glis3*. We demonstrated that circGlis3 is upregulated in the islets of genetic and dietary obese mouse models. In high-fat diet (HFD)-fed mice and older *Lepr*[db/db] mice, mimicking this increase resulted in enhanced insulin secretion and alleviated β-cell apoptosis. Mechanistically, circGlis3 enhances insulin transcription and secretion by sponging miR-124-3p and alleviates apoptosis in a Caspase 3 inhibition-dependent manner by binding the proapoptotic protein SCOTIN. Notably, in the presence of continuously elevated glycaemia, circGlis3 expression is upregulated by the splicing protein QKI, and then, the level of free circGlis3 is decreased by FUS, which accumulates in the cytoplasm when β-cells suffer excessive stress and decompensation. Collectively, circGlis3 is an obesity-responsive molecule and appears to be an essential regulator of β-cell biology. The upregulation of circGlis3 is related to the modification of the mass and function of β-cells during obesity.

## Results

### circGlis3 is elevated in the islets of obese male mice

We performed global circRNA expression profiling in pancreatic islets obtained from two mouse models of obesity—HFD-fed mice compared to normal chow diet (NCD)-fed mice and mice homozygous for the obesity-related mutation of leptin (*Lep*[ob/ob]) compared to wild-type littermates—to identify circRNAs potentially contributing to the development of obesity-associated disturbances in β-cell dysfunction and apoptosis.

These obese mouse models can compensate for insulin hypersecretion and β-cell mass for a long time[12]. Supplementary Fig. 1a-g shows the body weight, blood glucose, and insulin levels of these mice. 2738 and 2534 circRNAs were detected in the NCD/HFD and control/*Lep*[ob/ob] islets, respectively. There were 3646 diverse circRNAs in total. (Supplementary Fig. 1g and h). The expression of 302 circRNAs in the islets of the HFD-fed mice was significantly altered compared to that of the NCD controls, with 130 circRNAs increasing and 172 circRNAs decreasing (selection criteria: log2-fold change >2 or < -2, $p < 0.05$; Supplementary Fig. 1g. Source data are provided as a Source Data file.). In the *Lep*[ob/ob] islets, the expressions of 767 circRNAs varied considerably, with 413 circRNAs increasing and 354 decreasing (selection criteria: Log2 fold change >2 or < -2, $p < 0.05$; Supplementary Fig. 1h. Source data are provided as a Source Data file.). Stoll et al. previously detected annotated circRNAs in human islets, and conserved circRNAs in both human and mouse islets[13]. We compared our annotated circRNAs to conserved circRNAs from Stoll et al., and 95.6% of the circRNAs overlapped (supplied as Supplementary Data files). The raw RNA sequencing data of the HFD-fed mice islets was previously published[14] and uploaded to the Gene Expression Omnibus (GEO) database, with accession number GSE139991, and what is presented in this study is a reanalysis. The raw RNA-sequencing data of the *Lep*[ob/ob] mice islets was deposited in the Sequence Read Archive (SRA) database, with accession number SRR19137825 and SRR19137826. The reanalysis and analysis were both performed according to the circRNA data process flow.

The changes in the circRNA levels observed were confirmed by RT-PCR analysis of the 20 top dysregulated circRNAs: 10 upregulated and 10 downregulated circRNAs. Approximately 90% of the RT-PCR findings agreed with the RNA-seq results, indicating that the results of the RNA-seq data were credible (Supplementary Fig. 1i, j). We chose the annotated circRNA mmu_circRNA006170 (circBase ID: mmu_circ_0000943[15], termed circGlis3 in a subsequent study since its host gene is *Glis3*) for further study for the following reasons: (1) according to the

transcripts per kilobase million (TPM) in our RNA-seq data, circGlis3 was one of the most abundant circRNAs among the differentially expressed circRNAs (supplied as Supplementary Data files). (2) circGlis3 was enriched in the pancreas and was also detectable in the spleen, lung, and kidney, where its expression was more than 10 times lower than that in the pancreas (Fig. 1a). (3) circGlis3 was significantly upregulated in the pancreas of the HFD-fed mice, although the *mGlis3* (mRNA of *Glis3*) and GLIS3 protein levels were decreased (Fig. 1b, c), as previously reported[16]. These findings indicated that the higher expression of circGlis3 in obesity was more than just a byproduct of splicing and was suggestive of functionality.

Then, from 4 to 12 weeks of age, we measured the expression of circGlis3 in the islets of the aged *Lep*[ob/ob] mice and observed an increase in circGlis3 expression with the onset of insulin resistance at the beginning of 6 weeks of age (Fig. 1d). We also observed a similar increase in circGlis3 expression in the islets of the mice fed a 15-week HFD and the young *Lepr*[db/db] mice, while circGlis3 expression was significantly decreased, accompanied by glucose elevation (Fig. 1e, f), indicating that this observation is not limited to one mouse model of obesity and insulin resistance. The expression level of circGlis3 was also increased in other tissues of the obese mice compared to the normal mice, including the kidney, liver, lung, and brain (Supplementary Fig. 1k, l), but to a lesser extent than that observed in the islets. We also compared circGlis3 expression in primary islets versus exocrine glands in mice and found that islet expression was 4.5-fold higher than that in exocrine glands, indicating that circGlis3 was enriched in islets (Fig. 1g). Fluorescence in situ hybridization (FISH) analysis of circGlis3 in β cell, α cell and δ cell was performed in mouse primary islets and mouse pancreas slices. Furthermore, the results of β cell and α cell sorting and measurement of the level of circGlis3 confirmed that circGlis3 was especially enriched in β-cells (Supplementary Fig. 1m).

Furthermore, we quantified circGlis3 expression in the serum of a cohort of patients, and the results showed that subjects with impaired glucose tolerance (IGT) and/or T2DM had dynamic circGlis3 levels (Fig. 1h). After analysing the physiological indices of these subjects, we found an interesting phenomenon: the level of circGlis3 in human serum was significantly increased in individuals with obesity and moderate diabetes with compensated β-cell function, whereas the level of circGlis3 expression was decreased at the decompensation stage (Supplementary Fig. 1n). The clinical and biochemical characteristics of subjects who participated in the study are shown in Table 1. We then performed a correlation analysis of the upstroke until we reached a BMI of 30. The results showed that the level of circGlis3 was positively correlated with BMI ( < 30) (Supplementary Fig. 1o) and moderately correlated with HbA1c (Supplementary Fig. 1p) but weakly correlated with age. We next exposed human islets to different glucose concentrations for 48 h and detected the level of circGlis3, and the data were consistent with those in serum (Fig. 1i). Both studies demonstrated that serum levels might reflect the intracellular changes in the islet cells.

Generally, in dietary and genetic mouse models of obesity, as well as in individuals with IGT and/or T2DM, the level of circGlis3 in the islets was increased.

### The splicing factor QKI regulates the formation of circGlis3

Mouse circGlis3 was derived from exon3 (Chr19:28530873-28531986) of the *Glis3* gene (Gene ID: 226075; ENSMUSG00000052942) with a length of 1114 nt on chromosome 19 (Fig. 2a). The sequence matches the annotation in the circBase database (http://www.circbase.org/). Sequencing analysis confirmed the presence of backspliced junctions (Fig. 2b). There was more than 80% homology between human circGlis3 and mouse circGlis3 (Supplementary Fig. 2a). Divergent and convergent primers were designed for circGlis3 and linear transcripts. The cDNA and genomic DNA (derived from MIN6 cells) were amplified and analyzed using agarose gel electrophoresis (Fig. 2c). The stability

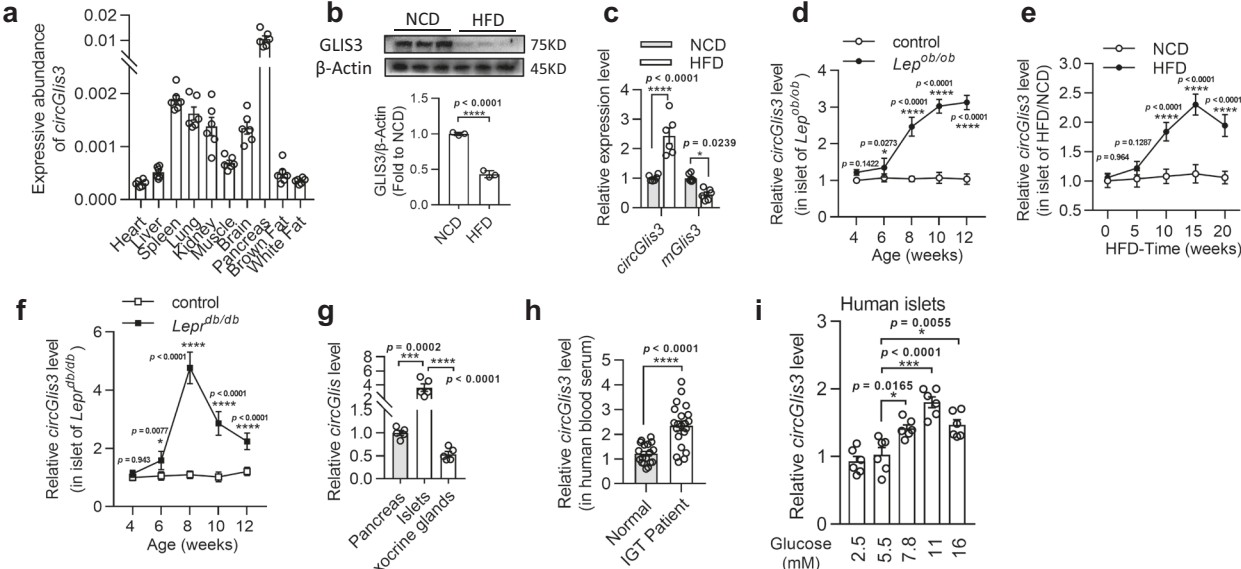

**Fig. 1 | circGlis3 is upregulated in and is associated with obesity. a** RT-PCR analysis of circGlis3 expressive abundance. Compared to other tissues, circGlis3 is enriched in pancreas (*n* = 6 biological animals). **b** Western Blotting showing GLIS3 protein level in pancreas of the normal diet (NCD)- and high fat diet (HFD)-fed mice. **c** RT-PCR analysis of circGlis3 and *Glis3* mRNA in pancreas of NCD- and HFD-fed mice (*n* = 6 biological animals). **d** RT-PCR analysis of circGlis3 expression in the islets of *Lep^ob/ob* mice aged from 4 to 12 weeks and HFD-fed mice from 0 to 20 weeks (**e**) and *Lepr^db/db* mice aged from 4 to 12 weeks (**f**). (*n* = 6 biological subjects). **g** circGlis3 expression level in mouse islets and exocrine glands (*n* = 6 biological subjects). **h**. circGlis3 expression level in the sera of normal and patients with impaired glucose tolerance (IGT) (Normal individual *n* = 18, patients with IGT *n* = 21). **i** circGlis3 expression level in human islets exposed to different concentrations of glucose for 48 h (*n* = 5-6 biological subjects). **a–i** For bar and line graphs, data represents mean ± SEM. **a**, **b**, **h** Unpaired two-tailed Student's *t*-test. **c–f** Two-way ANOVA with Bonferroni's post-test. **g**, **i** One-way ANOVA with Tukey's post-test. *$p < 0.05$, **$p < 0.01$, ***$p < 0.001$, ****$p < 0.0001$. Source data are provided as a Source data file.

and localization of circGlis3 in MIN6 cells were next investigated. Resistance to RNase R exonuclease digestion indicated that this RNA species is circular inform (Fig. 2d). The circular form of *Glis3* exon3 is preferentially localized in the cytoplasm, according to RT−PCR analyzes of nuclear and cytoplasmic circGlis3 and RNA in situ hybridization against circGlis3 (Fig. 2e, f).

The mechanism by which circGlis3 is formed was then explored. Previous research showed that splicing factors have a role in regulating circRNA biogenesis[17]. We postulated that if a protein factor contributes to circGlis3 biogenesis, it would also be regulated by obesity. We investigated the expression level of a candidate panel of splicing factors in the islets of the HFD-fed mice based on this hypothesis. We found that QKI increased in obese mice at both the mRNA and protein levels (Fig. 2g, h), and the *Qki* mRNA expression level was positively

correlated with that of circGlis3 in the islets of the HFD-fed mice (Fig. 2i, j). An identical trend in which the *Qki* mRNA expression level in islets was positively correlated with T2DM status, which involved a high fasting blood glucose level and long HFD-feeding time, was observed (Supplementary Fig. 2b, c). We also observed the same pattern in glucose- and palmitate-stimulated MIN6 cells (Supplementary Fig. 2d, e, Supplementary Fig. 2f, g). The effect of ectopic expression of QKI on circGlis3 formation was confirmed by RT−PCR analysis (Fig. 2k), and the transfection efficiency is shown in Supplementary Fig. 2h. QKI usually acts as a dimer capable of binding two well-separated regions of a single RNA molecule and promoting the proximity of circle-forming exons and circRNA biogenesis[18,19]. To determine whether QKI binding sites exist in the introns flanking the circRNA-forming exons of *Glis3*, we referenced the experimental scheme from Conn SJ[20] and searched for sequences that matched potential QKI response elements (QREs) in the vicinity of the QKI RNA-immunoprecipitation (RIP)-enriched regions. As previously reported, we obtained four instances of a bipartite motif that contains the sequence UAAY in conjunction with a relaxed version of the canonical QKI hexamer. As shown in Fig. 2l, two putative elements are positioned upstream, and two are positioned downstream of the circRNA-forming splice sites. We performed RIP assays and RT-PCR to quantify QKI occupancy within the introns adjacent to circGlis3-forming exon 3 to determine whether QKI binds *Glis3* pre-mRNA (Fig. 2m and Supplementary Fig. 2i). The pull-down assay established the binding interaction between the sites (which are in the introns flanking *Glis3* exon 3) and the QKI protein (Fig. 2n). Ectopic QKI expression in MIN6 cells, as well as additional RIP assays, was employed to evaluate whether QKI expression levels impacted the binding interaction between motif sites (QRE1&2, QRE3, QRE4) (Fig. 2o–q, Supplementary Fig. 2j–l). Subsequently, mutation of the single putative binding sites individually had little effect on circRNA formation, while mutation of both members of the upstream pair and the downstream pair substantially reduced circRNA formation, and mutation of all four sites was more effective (Fig. 2r). All these

## Table 1 | Demographic and clinical characteristics of patients

| | controls without IGT or T2DM (*n* = 29) | Patients with IGT or T2DM | |
| --- | --- | --- | --- |
| | | BMI < 30 (*n* = 75) | BMI ≥ 30 (*n* = 14) |
| Age (years) | 48.82 ± 8.69 | 50.24 ± 12.21 | 50.35 ± 8.97 |
| Sex (male/female), % | 20/9 | 43/32 | 9/5 |
| BMI (kg/m²) | 22.84 ± 2.09 | 24.89 ± 2.70 *** | 32.73 ± 2.75 ***, ### |
| HbA1c (%) | 5.40 ± 0.34 | 7.54 ± 2.28 *** | 8.82 ± 2.59 *** |
| C-peptide (nmol/L) | 0.81 ± 0.31 | 0.76 ± 0.40 | 0.83 ± 0.38 |
| Fastingplasma glucose (nmol/L) | 4.94 ± 0.36 | 9.11 ± 3.79 *** | 11.53 ± 6.40 *** |
| Relative circGlis3 level in serum | 1.25 ± 0.34 | 6.57 ± 4.49 *** | 1.45 ± 0.54 ### |

Data are mean ± SEM unless otherwise indicated. Symbols represent *p* values from Bonferroni adjustment for multiple comparisons: ***$p < 0.001$ compared with control subjects; ###$p < 0.001$ compared with Patients with IGT or T2DM (BMI < 30).

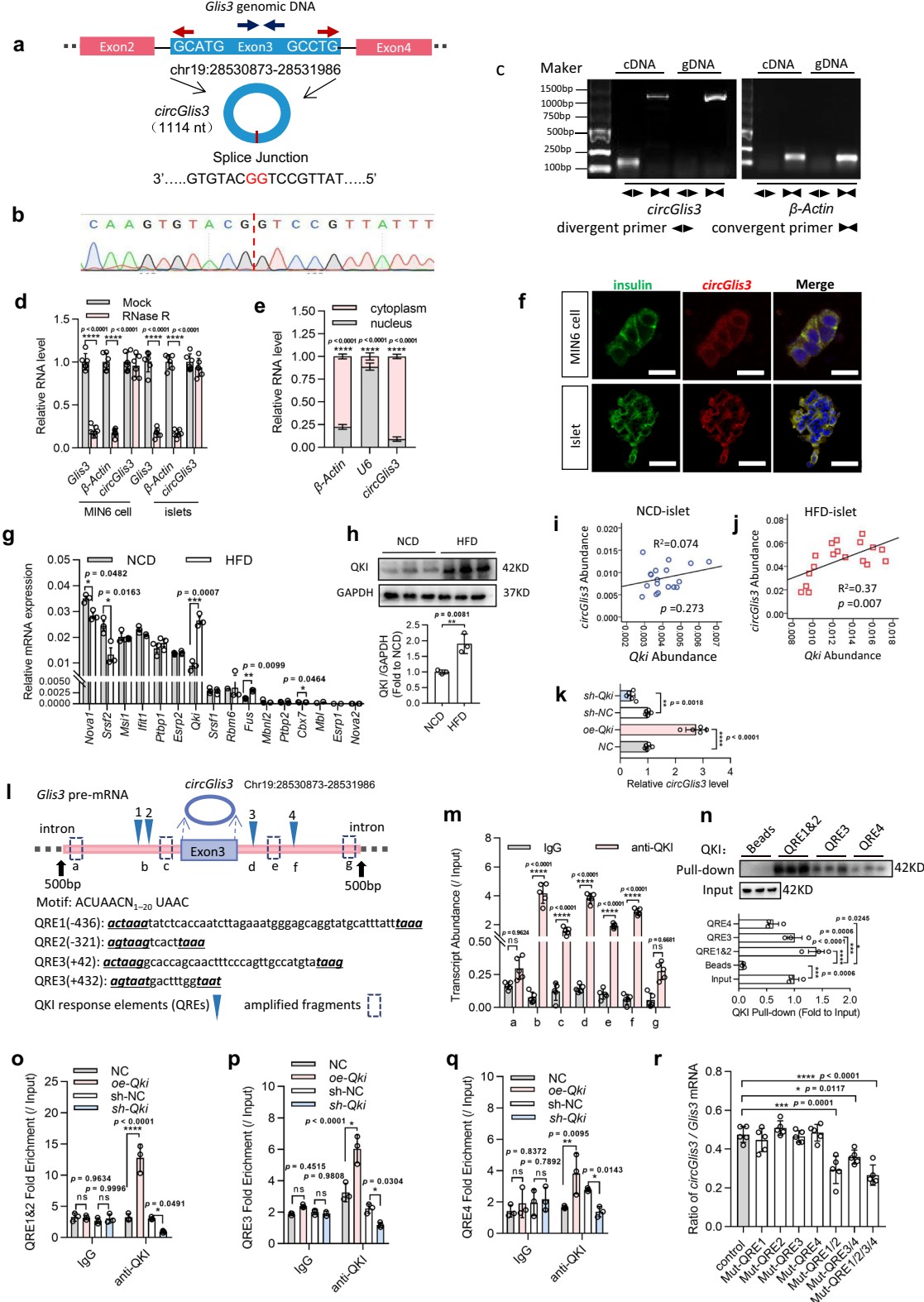

data indicated that QKI binds upstream and downstream of the circRNA-forming exon 3 in *Glis3* to promote circGlis3 formation.

## Upregulation of circGlis3 promotes insulin transcription and alleviates β-cell apoptosis in vitro

To explore the potential role of circGlis3 in regulating β-cell function, we used short hairpin RNA for knockdown of circGlis3 (sh-circGlis3), which is a group of three shRNAs to knock down circGlis3 in MIN6 cells

and primary islets, and a plasmid with a scramble sequence (*sh-NC*) was used as a control. In addition, an overexpression plasmid (oe-circGlis3) was used to upregulate circGlis3. Compared to the level of the linear transcripts, the level of the circular transcripts increased nearly 50-fold (Supplementary Fig. 3a, b). The results showed that the circularization was efficient. MIN6 cells and primary islets were transfected with sh-circGlis3, sh-NC, oe-circGlis3, or empty vector (pEx3ciR) and collected 48 h later. The efficiency of knockdown and overexpression

**Fig. 2 | Identification of circGlis3, and splicing factor QKI regulates formation of circGlis3. a** Circularization schematic of circGlis3 derived from exon3 of host gene *Glis3*. **b** Sequencing analysis of head-to-tail splicing junction in circGlis3. **c** circGlis3, along with *β-Actin*, were amplified from cDNA or gDNA from MIN6 cells with divergent and convergent primers by PCR assays, respectively (*n* = 3 independent experiments). **d** RT-PCR analysis of circGlis3 and *β-Actin* mRNA levels in MIN6 cells and primary islets with and without RNase R treatment (*n* = 6 biological replicates). **e** Subcellular fractionation of MIN6 cells and measurement of circGlis3 by RT-PCR analysis (*n* = 5 biological replicates). **f** RNA in situ hybridization against circGlis3 in MIN6 cells and islets (Red represents circGlis3, Green represents insulin, Blue represents DAPI). **g** Absolute quantification for various splicing factors in the islets of the NCD- and HFD-fed mice were measured by RT-PCR assays (*n* = 3 biological subjects). **h** Western Blots showing QKI protein in the islets of the NCD- and HFD-fed mice. **i, j** The correlation of circGlis3 and *Qki* mRNA abundance in the islets of the NCD- and HFD-fed mice (*n* = 18 biological animals). **k** The effect of ectopic expression of QKI on circGlis3 formation was confirmed by RT–PCR analysis (*n* = 6 biological replicates). **l** Schematic of *Glis3* pre-mRNA showing the locations of four putative QKI response elements (QREs) (inverted blue triangles) and amplicons (**a**–**g**) used for RNA-binding protein immunoprecipitation (RIP) assay, and sequence of QREs. Numbers in brackets refer to the distance from the circGlis3 forming splice site. **m** Fold enrichment of endogenous *Glis3* intron fragments by QKI in MIN6 cells were detected by RIP and RT-PCR, the RT-PCR primers were designed and indicated in Fig. 2l, *n* = 6 biological replicates. **n** Biotin-labeled *Glis3* intron fragments containing QREs were used for RNA-QKI pulldown against MIN6 cells lysate, and Western Blotting showing QKI was pulled down. **o**–**q** RIP and RT-PCR assays were used to measure the QRE1&2, QRE3, QRE4 fold enrichment (*n* = 3 biological replicates). **r** Effect of mutations to the QREs on the ratio of circRNA to linear mRNA, as determined by ratios of Absolute quantification by RT-PCR assays (*n* = 6 biological replicates). **d**–**e**, **g**–**k**, **m**–**r** For bar and line graphs, data represents mean ± SEM. **h** Unpaired two-tailed Student's *t*-test. **k, n, r** One-way ANOVA with Tukey's post-test. **d, e, g, m, o**–**q** Two-way ANOVA with Bonferroni's post-test. **i, j** Pearson correlation and regression analysis. $^*p < 0.05$, $^{**}p < 0.01$, $^{***}p < 0.001$, $^{****}p < 0.0001$. Source data are provided as a Source data file.

were approximately 60% and 40-fold, respectively (Supplementary Fig. 3c, d). oe-circGlis3 significantly increased the mRNA levels of the insulin genes (*Ins1* and *Ins2*) (Fig. 3a, b) and the insulin content (Fig. 3c, d), while knockdown of circGlis3 induced the opposite result. Then, we stimulated primary islets and MIN6 cells with circGlis3 knockdown or overexpression with glucose and palmitate. Suppression of circGlis3 expression decreased insulin secretion after exposure to high glucose (16.7 mM glucose) or palmitate (0.5 mM glucose), but overexpression of circGlis3 markedly increased insulin secretion (Fig. 3e, f; Supplementary Fig. 3e, f). However, circGlis3 did not affect glucagon and somatostatin secretion (Supplementary Fig. 3g, h). To explain how upregulation of circGlis3 promotes insulin production, we detected the mRNA expression of 42 transcription factors known to play important roles in insulin transcription and biosynthesis and β-cell maturation and function (Supplementary Fig. 3i, j). CircGlis3 overexpression strongly and selectively increased CREB1 and NeuroD1 at both the mRNA and protein levels (Fig. 3g, h), and the results also provided a basis for subsequent related research. These results suggested that the processes responsible for the increase in insulin content occurred at the transcription level.

To identify the effect of circGlis3 on the maintenance of β-cell mass, we modified circGlis3 expression in MIN6 cells. The CCK-8 assay showed that regulating circGlis3 had no discernable effect on cell proliferation (Supplementary Fig. 3k), and Ki67 immunofluorescence staining also confirmed this result (Supplementary Fig. 3l). circGlis3 overexpression in MIN6 cells resulted in a striking reduction in palmitate-induced apoptosis, as showed by Annexin V/PI staining and quantification of the cells displaying nuclei, suggesting that circGlis3 upregulation results in resistance to obesity-induced apoptosis of β-cells (Fig. 3j). The TUNEL assay was then performed in MIN6 cells and mouse islets, as well as in human islets, and the results showed that downregulation of circGlis3 increased the number of apoptotic cells, whereas overexpression of circGlis3 resulted in the opposite outcomes (Fig. 3k–m). Furthermore, Western Blotting indicated that apoptotic proteins were altered in abnormal circGlis3-treated cells, consistent with the flow cytometric and TUNEL analysis. Suppression of circGlis3 expression induced cleaved-Caspase 3 and BAX expression while decreasing BCL-2 expression, and overexpression induced the opposite result (Fig. 3i). These findings indicated that the upregulation of circGlis3 can alleviate β-cell apoptosis during obesity.

### Overexpression of circGlis3 protects against β-cell dysfunction and apoptosis in vivo

To test whether overexpression of circGlis3 can alleviate obesity-induced β-cell dysfunction in vivo, we injected $1 \times 10^{12}$ adeno-associated virus particles expressing circGlis3 (oe-circGlis3) into male C57BL/6 mice aged 8 weeks via pancreatic ductal infusion, and the mice were subsequently fed an HFD for at least 16 weeks. Figure 4a depicts the time pattern of oe-circGlis3-vector injection and HFD feeding. The circularization efficiency in vivo was also assessed by RT-PCR (Supplementary Fig. 4a, b). Even 20 weeks after injection, we observed a 9-fold increase in islet circGlis3 expression in the mice given the oe-circGlis3 compared to those receiving adeno-associated virus particles with an empty vector (oe-vector) (Supplementary Fig. 4c). In other organs, the expression level of circGlis3 was not significantly altered (Supplementary Fig. 4d). The oe-circGlis3 treatment had no impact on cumulative energy intake (Supplementary Fig. 4e) or body weight (Supplementary Fig. 4f). This treatment did not influence fasting glucagon, although it was slightly lower than that in the control mice (Supplementary Fig. 4g). Furthermore, we isolated islets from the oe-circGlis3-treated mice and measured somatostatin secretion in response to 16.7 mM glucose, and oe-circGlis3 did not affect somatostatin secretion (Supplementary Fig. 4h). We discovered that pancreatic ductal infusion did not affect α cells or δ cells. After 3 months of HFD feeding, there was no significant difference in fasting blood glucose between the control and oe-circGlis3-treated mice (Supplementary Fig. 4i). However, the homeostatic model assessment of insulin resistance (HOMA-IR) indices of oe-circGlis3-treated mice were significantly decreased (Fig. 4b). Then, we found that proinsulin-to-insulin ratio values were obviously restrained in the oe-circGlis3-treated HFD-fed mice, although both insulin levels and proinsulin levels were higher than those in the control HFD-fed mice (Fig. 4c–e). The results confirmed that overexpression of circGlis3 improved β-cell biological function. Furthermore, glucose tolerance tests (IPGTT) revealed that circGlis3 overexpression improved glucose tolerance (Fig. 4f). The insulin tolerance tests (IPITT) suggested that circGlis3 upregulation improved insulin sensitivity (Fig. 4g). In addition, the in vivo glucose-stimulated insulin secretion (GSIS) data suggested that oe-circGlis3 treatment enhanced insulin secretion (Fig. 4h). Furthermore, after 8 and 16 weeks of HFD feeding, we isolated islets from the oe-circGlis3-treated and control mice and performed GSIS tests. Insulin release was markedly improved in the oe-circGlis3-treated mice when islets were exposed to 16.7 mM glucose, according to the findings (Fig. 4i, j). Using morphometric analysis of the pancreatic sections, we found that β-cell mass was 1.7-fold higher in the oe-circGlis3-treated mice than in the control mice (Fig. 4k). TUNEL assays further revealed that apoptosis-positive β-cell counts were reduced by 67% and 56% in the oe-circGlis3-treated mice after 8 and 20 weeks of HFD feeding, respectively, compared to those in the control animals (Fig. 4l). Corresponding to the above results, the upregulation of circGlis3 resulted in a significant decrease in cleaved-Caspase 3 and BAX expression while increasing BCL-2 expression (Fig. 4m).

To confirm the effect of circGlis3 on β-cell protection, we injected the identical virus particles into 4-week-old *Lepr*$^{db/db}$ mice. We found

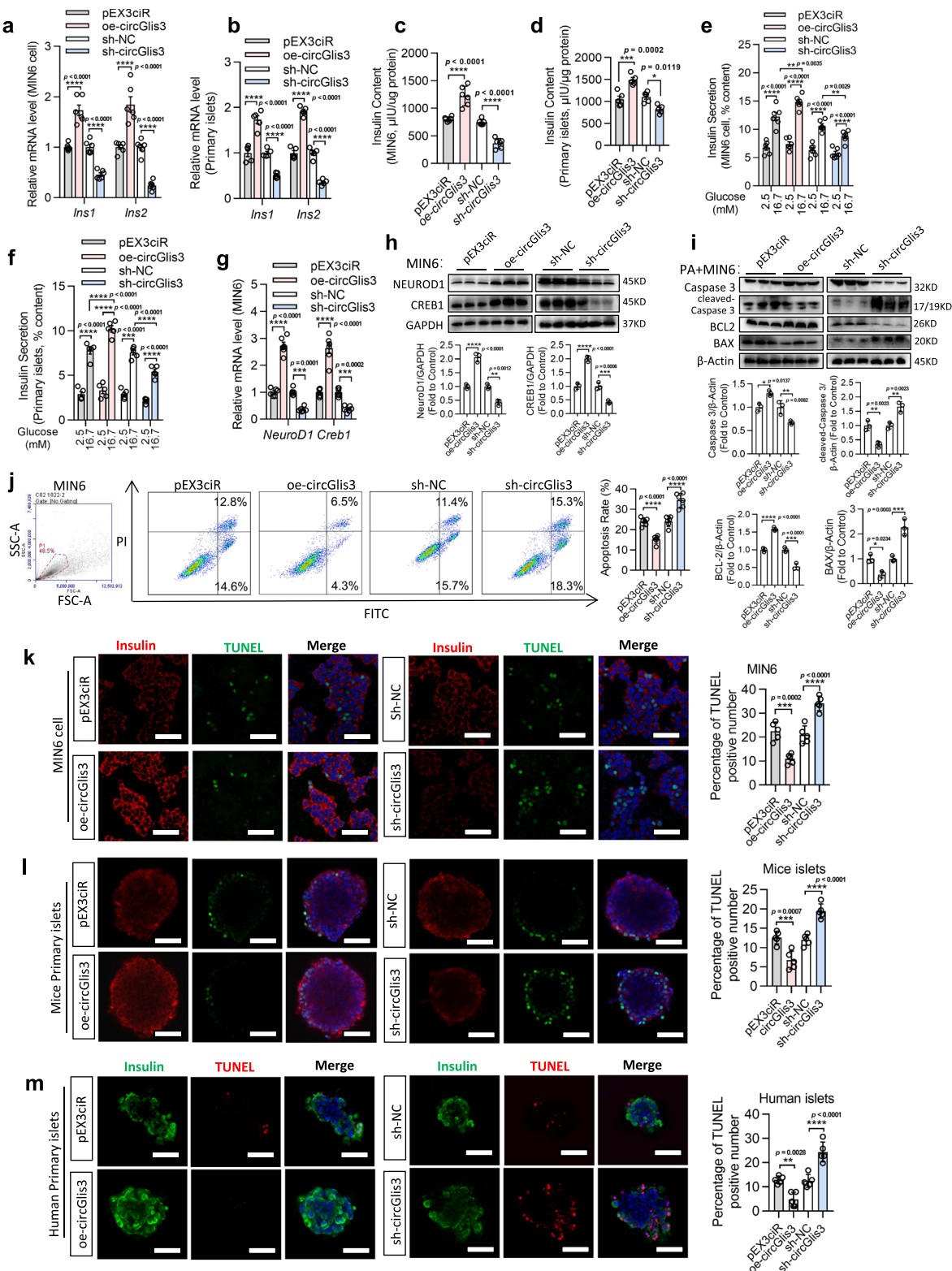

that overexpression of circGlis3 reduced blood glucose from 10 weeks to 14 weeks (Supplementary Fig. 4j) and restrained the proinsulin-to-insulin ratio values while increasing the insulin and proinsulin levels (Fig. 4n–o). Then, we performed in vivo IPITT and GSIS assays, and the results showed that insulin resistance was decreased in the oe-circGlis3-treated *Lepr^db/db* mice (Fig. 4p) and that oe-circGlis3 treatment enhanced insulin secretion (Fig. 4q). Further investigations demonstrated that the overexpression of circGlis3 in the *Lepr^db/db* mice maintained the β-cell mass (Fig. 4s) while decreasing the number of

TUNEL-positive β-cells (Fig. 4r). These results indicated that over-expression of circGlis3 in dietary and genetic mouse models of obesity improves β-cell function while inhibiting β-cell apoptosis. The above results confirmed that the effect of circGlis3 on glucose homeostasis is β-cell-mediated.

**circGlis3 regulates insulin transcription by sponging miR-124-3p**
circRNAs may function as competing endogenous RNAs (ceRNAs) to sponge miRNAs, thereby modulating miRNA target depression and

**Fig. 3 | Upregulation of circGlis3 promotes insulin transcription and secretion, and inhibits β-cell apoptosis in vitro. a** Expression of *Ins2* and *Ins2* mRNA in MIN6 cells (*n* = 6 biological replicates) and mouse islets (*n* = 5 biological subjects) with circGlis3 overexpression and knockdown. **c, d** Insulin content in MIN6 cells (*n* = 6 biological replicates) and mouse islets (*n* = 5 biological subjects) with circGlis3 overexpression and knockdown. **e, f** Insulin secretion in glucose (2.5 mM and 16.7 mM) stimulated MIN6 cells (*n* = 6 biological replicates) and mouse islets (*n* = 5 biological subjects) with circGlis3 overexpression and knockdown. **g, h** *NeuroD1* and *Creb1* mRNA and protein levels in MIN6 cells with circGlis3 overexpression and knockdown (*n* = 6 biological replicates). **i** Western Blotting showing Pro-CASPASE 3, Cleaved-CASPASE 3, BAX, and BCL-2 proteins in MIN6 cells with circGlis3 overexpression and knockdown. **j** Annexin V/PI staining and flow cytometry analysis of cell apoptosis in MIN6 cells with circGlis3 overexpression and knockdown (*n* = 5 biological replicates). **k–m** TUNEL assays and statistic results of cell apoptosis in MIN6 cells (Red represents Insulin, Green represents positive TUNEL cells; Scale bars represent 50 μm), mouse islets (Red represents Insulin, Green represents positive TUNEL cells; Scale bars represent 100 μm), and human islets (Red represents positive TUNEL cells, Green represents Insulin; Scale bars represent 100 μm) with circGlis3 overexpression and knockdown (*n* = 5-6 biological replicates). **a–m** For bar graphs, data represents mean ± SEM. **a, b, e–g** Two-way ANOVA with Bonferroni's post-test. **c, d, h–m** One-way ANOVA with Tukey's post-test. $^{*}p < 0.05$, $^{**}p < 0.01$, $^{***}p < 0.001$, $^{****}p < 0.0001$. Source data are provided as a Source data file.

imposing an additional level of post-transcriptional regulation[21]. To identify the potential miRNA targets of circGlis3, we performed an in silico analysis using the StarBase, miRanda, RNAhybrid and TargetScan databases, which collectively predicted that 4 miRNAs may be biological targets of *circGlis3* (Fig. 5a). According to the ceRNA theory, the expression of circRNA and miRNA should have a negative correlation[22]. The expression patterns of the aforementioned miRNAs were next examined in the islets of the HFD-fed mice and *Lep^{ob/ob}* mice. Among the target miRNAs, miR-124-3p and miR-298-5p were shown to be elevated in the islets of obese mice (Fig. 5b, c). The direct binding of the high abundance miRNAs (miR-124-3p and miR-298-5p) to circGlis3 was showed by affinity pull-down of endogenous miRNAs associated with circGlis3 utilizing in vitro transcribed biotin-labeled circGlis3 sense or anti-sense and confirmed via RT-PCR analysis. First, the binding efficiency of biotin-labeled circGlis3 to beads was detected (Supplementary Fig. 5a). Compared to the negative control (beads) and biotin-labeled circGlis3 anti-sense, the biotin-labeled circGlis3 sense significantly captured endogenous miR-124-3p in MIN6 cells (Fig. 5d). However, there was a modest enrichment for miR-298-5p (Supplementary Fig. 5b). Moreover, a RNA pulldown assay using biotin-labeled miR-124-3p was performed. The results revealed that miR-124-3p captured endogenous circGlis3; however, the negative control with a disrupting putative binding sequence failed to coprecipitate out circGlis3 (Fig. 5e). We also conducted the RNA in situ hybridization assay and discovered that miR-124-3p and circGlis3 were colocalized in the cytoplasm (Fig. 5f). The above data indicated that circGlis3 colocalized with miR-124-3p in the cytoplasm.

Supplementary Fig. 5C illustrates the predicted binding sites of miR-124-3p to circGlis3. We constructed a dual-luciferase reporter by inserting the wild-type (WT) or mutant (MT) linear sequence of circGlis3 into the pMIR-REPORT luciferase vector. Overexpression of miR-124-3p reduced the luciferase activities of the WT reporter vector but not the mutant reporter vector (Fig. 5g). Upregulation of circGlis3 reduced the level of miR-124-3p in both MIN6 cells and mouse islets, as predicted (Fig. 5h). However, there was no significant difference in circGlis3 levels when miR-124-3p mimics or inhibitors were transfected into MIN6 cells or primary islets (Supplementary Fig. 5d). To determine whether circGlis3 affects the synthesis of miR-124-3p, we measured the levels of pri-miR-124-3p and pre-miR-124-3p by RT-PCR assays. We found that overexpression of circGlis3 did not affect the transcription of miR-124-3p (Supplementary Fig. 5e, f). This result indicated that miR-124-3p was bound to circGlis3 but did not induce its expression. All of these findings demonstrated that circGlis3 functioned as a molecular sponge for miR-124-3p.

Related rescue experiments were performed to investigate whether circGlis3 regulates β-cell function by sponging miR-124-3p. miR-124-3p was shown to function in insulin transcription and secretion. Rescue experiment results showed that the decrease in insulin transcription caused by overexpression of miR-124-3p was partially reversed by overexpression of circGlis3 (Fig. 5i–k, Supplementary Fig. 5g). The inhibition of insulin secretion in MIN6 cells (Fig. 5l), mouse islets (Fig. 5m) and human islets (Fig. 5n) which were mediated by increasing miR-124-3p were likewise abolished by overexpression of circGlis3. Furthermore, in the oe-miR-124-3p-treated HFD-fed mice, the area under the curve obtained from IPGTT assays was increased (Fig. 5o); the area under the curve obtained from glucose-stimulated insulin secretion assays was decreased (Fig. 5p), and these effects were markedly ameliorated by oe-circGlis3 treatment.

We subsequently investigated whether circGlis3-associated miR-124-3p regulates insulin-related gene expression by targeting the 3′-untranslated region (UTR). Prediction of target genes of miR-124-3p was performed by TargetScan, StarBase, miRDB, and miRWalk. *Creb1* and *NeuroD1*, which are as key insulin-specific transcription factors, were shown to be at the intersection of the predictions (Supplementary Fig. 5h). The predicted binding sites of miR-124-3p to *Creb1* and *NeuroD1* are illustrated in Supplementary Fig. 5i, j. The luciferase reporter assay was applied to verify the targeting ability through the pMIR-REPORT vector, which included either the WT or MUT 3′-UTR of *NeuroD1* and *Creb1*, respectively. Overexpression of miR-124-3p reduced the luciferase activities of the WT reporter vector but not the MUT reporter vector (Fig. 5q, r). The ability of miR-124-3p to regulate the *NeuroD1* and *Creb1* mRNA and protein levels in MIN6 cells was next studied. When MIN6 cells were transfected with miR-124-3p mimics, the mRNA and protein levels of *NeuroD1* and *Creb1* were significantly decreased (Supplementary Fig. 5k, l). *NeuroD1* and *Creb1* mRNA and protein levels were increased in the oe-circGlis3-treated MIN6 cells (Fig. 3g, h) and mice (Supplementary Fig. 5m, n). The rescue experiment results showed that the overexpression of circGlis3 reversed the inhibitory effect of miR-124-3p on the mRNA and protein expression of *Creb1* and *NeuroD1* in vitro (Fig. 5s, t) and in vivo (Fig. 5u, v). The findings indicated that circGlis3 functions as a sponge for miR-124-3p and upregulates the expression of *NeuroD1* and *Creb1*, resulting in promotion of insulin transcription and secretion.

## circGlis3 interacts with SCOTIN to prevent β-cell apoptosis

The mechanisms by which circGlis3 prevented β-cell apoptosis during obesity were then examined. CircRNA can regulate the functionality of RNA-binding proteins by direct binding in addition to sponging miRNAs[23,24]. To investigate this possibility for circGlis3, biotinylated circGlis3 was prepared through assays using in vitro transcription and cyclization reactions (Supplementary Fig. 6a and b), pulled it down with anti-biotin beads, and subjected the pulled-down samples to mass spectrometry. Numerous proteins were identified as potent circGlis3-interacting proteins from this analysis (supplied as Supplementary Data files). SCOTIN and FUS were two of the top ten candidates identified (Fig. 6a, Supplementary Fig. 6c and d). SCOTIN is a proapoptotic factor that is activated in a Caspase-dependent manner in response to DNA damage or cellular stress[25]. We hypothesized that circGlis3 may arrest β-cell apoptosis by modulating SCOTIN.

We further confirmed the interaction between circGlis3 and SCOTIN by performing additional independent experiments. First, we performed RIP assays, which confirmed that endogenous circGlis3 interacts with SCOTIN in MIN6 cells (Fig. 6b). Second, we utilized biotin-labeled circGlis3 to pull down endogenous SCOTIN and used Western Blot analysis to identify the SCOTIN protein (Fig. 6c). Finally, in MIN6 cells and islets, we coupled the circFISH protocol for circGlis3

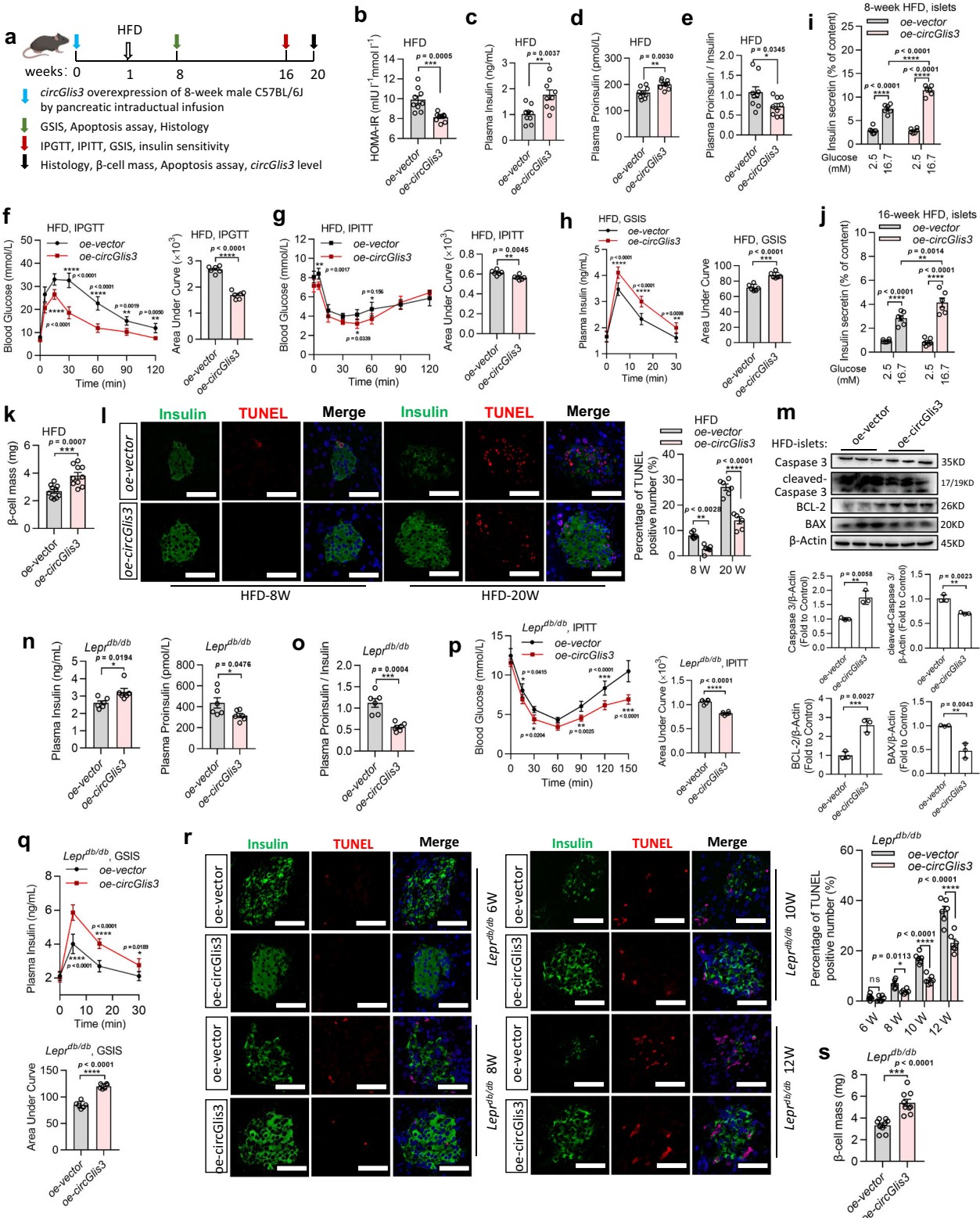

visualization with immunofluorescence staining for SCOTIN, and the results demonstrated that circGlis3 colocalized with SCOTIN in the cytoplasm (Fig. 6d). These complementary experiments consistently demonstrated that endogenous circGlis3 can interact with SCOTIN. These interactions occur mainly in the cytoplasm, where SCOTIN mostly resides. Based on these findings, we then aimed to investigate whether circGlis3 functions by regulating SCOTIN. We found that SCOTIN was increased in the islets of the 20-week HFD-fed and $Lepr^{db/db}$

mice (Fig. 6e, f), as well as in the MIN6 cells incubated with palmitate (Supplementary Fig. 6e, f) and glucose (Supplementary Fig. 6g, h), indicating increased expression of SCOTIN under lipotoxic conditions. However, circGlis3 ectopic expression did not affect SCOTIN transcription and translation (Supplementary Fig. 6i, j).

The rescue experiments, as assessed by Annexin V/PI staining, demonstrated that SCOTIN overexpression in the palmitate-treated MIN6 cells resulted in increased apoptosis, but upregulation of

**Fig. 4 | Overexpression of circGlis3 protects against β-cell dysfunction and apoptosis in vivo. a** The time pattern of oe-circGlis3-vector injection and HFD feeding (*n* = 12 biological animals). **b** homeostasis model assessment of insulin resistance (HOMA-IR) of the negative control and oe-circGlis3-treated HFD-fed mice (*n* = 9 biological animals). HOMA-IR was calculated as HOMA-IR = (FBG (mmol/L)×FINS (mIU/L))/22.5. **c–e** The fasting insulin, proinsulin and proinsulin-to-insulin ratio values in fasting serum levels of the negative control and oe-circGlis3-treated HFD-fed mice (*n* = 9 biological animals). **f, g** Intraperitoneal glucose tolerance test (IPGTT) (2 g/kg) and intraperitoneal insulin tolerance test (IPITT) (1 U/kg) were performed in the overnight fasted HFD-fed mice (*n* = 6 biological animals). The corresponding area under the curve (AUC) was calculated. **h** in vivo glucose-stimulated insulin secretion (GSIS) were performed in the overnight fasted HFD-fed mice (*n* = 9 biological animals). The corresponding area under the curve (AUC) was calculated. **i, j** Insulin secretion in glucose (2.5 mM and 16.7 mM) stimulated islets from mice with HFD-fed for 8 weeks and 16 weeks (*n* = 6 biological subjects). **k** β-cell mass in the negative control and oe-circGlis3-treated HFD-fed mice (*n* = 10 biological animals). **l** TUNEL assays of pancreatic sections from negative control and oe-circGlis3-treated mice with HFD-fed for 8 weeks and 20 weeks (Red represents

positive TUNEL cells, Green represents Insulin; Scale bars represent 100 μm). **m** Western Blotting showing Pro-Caspase 3, Cleaved-CASPASE 3, BAX, and BCL-2 proteins in islets of the negative control and oe-circGlis3-treated HFD-fed mice. **n, o** The fasting insulin, proinsulin and proinsulin-to-insulin ratio values in fasting serum levels of the negative control and oe-circGlis3-treated *Lepr^{db/db}* mice (*n* = 6 biological animals). **p** IPITT (1.5 U/kg) performed in overnight fasted *Lepr^{db/db}* mice (*n* = 6 biological animals). The corresponding area under the curve (AUC) was calculated. **q** in vivo GSIS (1.5 g/kg body weight) were performed in overnight fasted *Lepr^{db/db}* mice (*n* = 6-9 biological animals). The corresponding area under the curve (AUC) was calculated. **r** TUNEL assays of pancreatic sections from the negative control and oe-circGlis3-treated *Lepr^{db/db}* mice at age from 6 to 12 weeks (Red represents positive TUNEL cells, Green represents Insulin; Scale bars represent 50 μm). **s** β-cell mass in the negative control and oe-circGlis3-treated *Lepr^{db/db}* mice (*n* = 9 biological animals). **b–s** For bar and line graphs, data represents mean ± SEM. **b–e, k, m–o, s** Unpaired two-tailed Student's *t*-test. **i, j, l, r** Two-way ANOVA with Bonferroni's post-test. **f–h, p–q** One-way ANOVA with Tukey's post-test and Unpaired two-tailed Student's *t*-test. $^*p < 0.05$, $^{**}p < 0.01$, $^{***}p < 0.001$, $^{****}p < 0.001$. Source data are provided as a Source data file.

---

circGlis3 abrogated this effect (Fig. 6g). TUNEL assays were next performed in the palmitate-treated MIN6 cells, mouse islets, and human islets, and the results showed that the upregulation of SCOTIN increased the number of positive-apoptotic cells, and this effect could be partially reversed by circGlis3 overexpression (Fig. 6h, i). Further rescue experiment results in vivo demonstrated that compared to that of the control HFD-fed mice, the largest decrease in β-cell mass caused by oe-SCOTIN treatment was partially reversed by oe-circGlis3 treatment (Fig. 6j). Further rescue experiments via TUNEL assays also showed that apoptosis-positive β-cell counts were markedly increased in the oe-SCOTIN-treated mice after 20 weeks of HFD feeding but were rescued by oe-circGlis3 treatment (Fig. 6k). These results suggested that upregulation of SCOTIN results in resistance to the obesity-induced apoptosis of β-cells; however, upregulation of circGlis3 abrogated this effect. SCOTIN usually acts as an endoplasmic reticulum (ER) membrane protein, and SCOTIN has been shown to induce apoptosis in a Caspase-dependent manner[26]. The oe-circGlis3 treatment inhibited the upregulation of cleaved-Caspase 3 protein expression produced by oe-SCOTIN in vitro (Fig. 6l) and in vivo (Fig. 6m). These results indicated that circGlis3 prevents β-cell apoptosis in a Caspase-dependent manner during obesity by interacting with SCOTIN and restraining the activity of Caspase 3.

### FUS sequesters circGlis3 to reduce its abundance in diabetes

Finally, we investigated why the expression level of circGlis3 was reduced during diabetes and lost its protective effect on β-cells. QKI regulates the production of circGlis3, as previously stated. However, the expression level of QKI did not decrease prominently with the occurrence of diabetes (Fig. 7a, b). This finding indicated that additional mechanisms may regulate the level of circGlis3 in diabetes. Using a circGlis3 pulldown assay and mass spectrometry analysis, we found that FUS was pulled down by biotin-labeled circGlis3 sense but not by anti-sense (Supplementary Fig. 7a). FUS is a 70 kDa RNA-binding protein with multiple functions that may shuttle between the nucleus and cytoplasm in response to stress[27]. Interestingly, we discovered that the FUS mRNA and protein levels were barely increased in islets from the HFD-fed mice (Fig. 7c, d) and MIN6 cells stimulated by glucose (Supplementary Fig. 7b, c) and palmitate (Supplementary Fig. 7d, e). Consistent with earlier studies[28], FUS was mostly observed in the cytoplasm after stress (Fig. 7e). Western blot analysis with an anti-FUS antibody indicated the existence of FUS within the circGlis3 sense pulldown samples from MIN6 cells (Fig. 7f). Moreover, a RIP assay with the FUS antibody showed that endogenous FUS directly bound to circGlis3 in MIN6 cells (Fig. 7g). But ectopic expression of circGlis3 had no direct effect on FUS transcription or translation (Supplementary Fig. 7f, g). However, we observed that overexpression of *FUS* reduced the free

copies of endogenous circGlis3 in MIN6 cells and islets (Fig. 7h), as well as the efficiency of FUS overexpression (Supplementary Fig. 7h).

We designed a series of competitive inhibition experiments to verify the regulatory effect of FUS on circGlis3. We found that overexpression of FUS accompanied by palmitate stimulation might reverse the inhibitory effect of circGlis3 on miR-124-3p, as evidenced by an increase in miR-124-3p and a decrease in its target genes (Fig. 7i). Similarly, the combination of FUS and SCOTIN with circGlis3 is competitive. The results of SCOTIN-RIP showed that FUS overexpression could competitively inhibit the enrichment of circGlis3 by SCOTIN in the presence of palmitate (Fig. 7j), followed by an increase in TUNEL-positive β-cells in the HFD-fed mouse islets (Supplementary Fig. 7i). Conversely, after overexpression of SCOTIN, FUS combined with circGlis3 also decreased (Fig. 7k).

During cellular stress, the nuclear protein FUS quickly assembles cytoplasmic stress granules (SG). Zhang's results supported the hypothesis that, in islet cells, lipotoxicity disrupts nucleocytoplasmic transport by inducing SG formation[29]. In a study by Khong, et al[30], SG was shown to restrict the diffusion of mRNAs and ncRNAs. We then stained circGlis3 and FUS in palmitate-stimulated MIN6 cells and pancreatic tissue from the HFD-fed mice and discovered that compared to normal conditions, lipotoxicity can cause dispersed distribution of FUS in the cytoplasm and nucleus with abundant SG assembly in the palmitate-stimulated MIN6 cells (white arrows) (Fig. 7l) and islets from the HFD-fed mice (Fig. 7m). The addition of emetine (10 μg/mL) reduced SG assembly, which markedly attenuated the cytoplasmic redistribution of FUS in the presence of cellular stress while not affecting the total protein level[31]. We stained circGlis3 and FUS in the palmitate-stimulated MIN6 cells with *si-FUS* or emetine (10 μg/mL) treatment, to explore whether the reduced circGlis3 is recruited into the FUS-SG assembly. The results showed that the decreasing number of FUS-formed SG rescued circGlis3 expression in the cytoplasm (Fig. 7l). RT-PCR results revealed a similar change in circGlis3 expression (Fig. 7n). The incidents illustrated that, during the development of diabetes, FUS competes with miR-124-3p and SCOTIN to bind circGlis3, resulting in a decrease in circGlis3 by restricting diffusion in the cytoplasm via the recruitment function of FUS-formed SG.

## Discussion

β-cell adaptation is a major mechanism in preventing T2DM progression, and failed β-cell compensation predicts the onset of T2DM[32]. This chronic progressive pathological process of T2DM indicates that there is an extended period between functional insulin depletion in β-cells and β-cell demise[33]. The molecular basis underlying compensatory β-cell mass remains unknown. In rodents, we discovered that β-cell

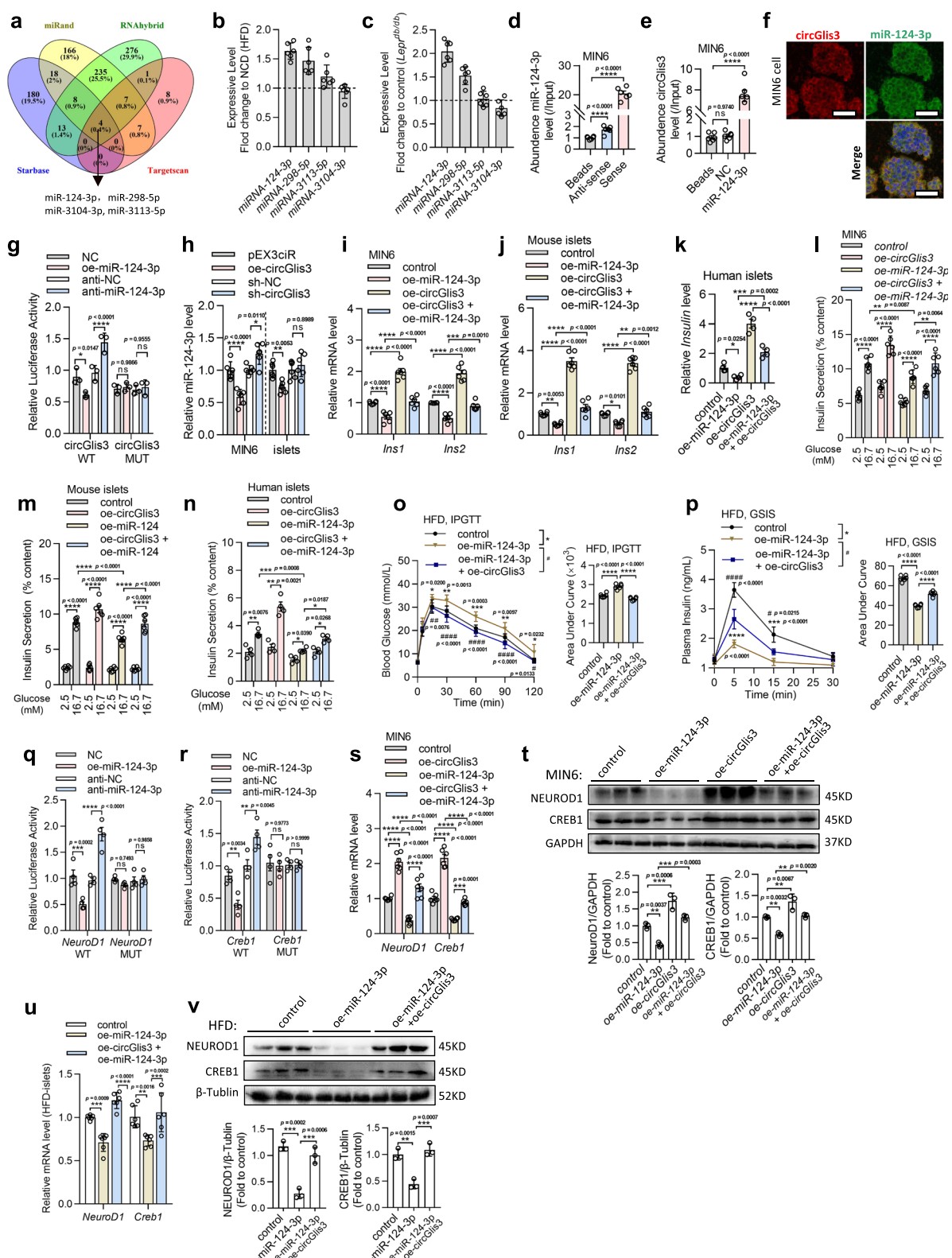

mass expansion during obesity is associated with increased expression of circGlis3. By activating QKI splicing in isolated pancreatic islets and MIN6 cells, we were able to recapitulate the increased circGlis3 level observed in obese mouse models. When diabetes arises, circGlis3 is elevated in the islets of obese mice models, which protects against islet β-cell dysfunction and apoptosis, while it decreases via FUS-formed SG (Fig. 8). These findings suggested that circGlis3 plays an important role in compensatory β-cell function and mass during obesity.

We investigated the possible causes of the changes in circGlis3 expression in the islets of individuals with obesity and found that the splicing factor QKI regulates circularization, which contributes substantially to the regulation of circGlis3 production. QKI, which belongs to the STAR family of KH domain-containing RNA-binding proteins, has been shown to regulate pre-mRNA splicing[34] and has been implicated in various diseases including diabetes. QKI can bind bipartite sequence motifs[35] on the same or separate RNA molecules and

**Fig. 5 | circGlis3 regulates insulin transcription by targeting and sponging miR-124-3p. a** Schematic illustration showing the overlap of the target miRNAs of circGlis3 predicted by StarBase, miRanda, RNAhybrid and TargetScan databases. **b, c** The relative levels of four miRNA candidates in islets from HFD-fed mice and *Lep^{ob/ob}* mice were detected by RT-PCR (*n* = 6 biological animals). **d.** RNA pulldown and RT-qPCR assay showing affinity pulldown of endogenous miRNAs by biotin-labeled circGlis3 sense or anti-sense (*n* = 6 biological replicates). **e** RNA pulldown and RT-PCR assay showing the amount of endogenous circGlis3 pulled down by biotin-labeled miR-124-3p (*n* = 6 biological replicates). **f** miR-124-3p and circGlis3 were colocalized in the cytoplasm (Red represents circGlis3, Green represents miR-124-3p, Scale bars represent 20 μm). **g** circGlis3 wild type (WT) and circGlis3 mutant type (MUT) sequence were cloned into the 3′-UTR of the pMIR-REPORT Luciferase reporter, and transfection miR-124-3p mimic repressed the reporter activity of pMIR-circGlis3-WT, and such repression could be restored by anti-miR-124-3p (*n* = 3 biological replicates). **h** RT-PCR analysis of miR-124-3p expression in MIN6 cells (*n* = 6 biological replicates) and islets (*n* = 6 biological subjects) with circGlis3 overexpression and knockdown. **i, j** RT-PCR analysis of *Ins1* and *Ins2* mRNA expression in the MIN6 cells (*n* = 6 biological replicates) and islets with oe-miR-124-3p or/and oe-circGlis3 treatment (*n* = 6 biological subjects). **k** RT-PCR analysis of *Insulin* gene level in human islets (*n* = 4 biological subjects) with oe-miR-124-3p or/and oe-circGlis3 treatment. **l, m** Insulin secretion in MIN6 cells (*n* = 6 biological replicates) and islets (*n* = 6 biological subjects) with oe-miR-124-3p or/and oe-circGlis3 treatment. **n** Insulin secretion in human islets with oe-miR-124-3p or/and oe-circGlis3 treatment (*n* = 4 biological subjects). **o, p** IPGTT and in vivo GSIS were performed in the overnight fasted HFD-fed mice with oe-miR-124-3p or/and oe-circGlis3 treatment (*n* = 6 biological animals). The corresponding AUC was calculated. **q, r** Relative luciferase activity of the pMIR-REPORT constructs containing either the WT or MUT 3′-UTR of the *NeuroD1* and *Creb1* (*n* = 3 biological replicates). **s, t** *NeuroD1* and *Creb1* mRNA and protein expression levels in MIN6 cells with oe-miR-124-3p or/and oe-circGlis3 treatment (*n* = 6 biological replicates). **u, v.** *NeuroD1* and *Creb1* mRNA and protein expression levels in the islets of the HFD-fed mice with oe-miR-124-3p or/and oe-circGlis3 treatment (*n* = 6 biological subjects). **b–e, g–v** For bar and line graphs, data represents mean ± SEM. **b–e, h, k, t, v** One-way ANOVA with Tukey's post-test. **g, i–j, l–n, q–s, u** Two-way ANOVA with Bonferroni's post-test. **o–p** One-way ANOVA with Tukey's post-test and Unpaired two-tailed Student's t-test. *p < 0.05, **p < 0.01, ***p < 0.001, ****p < 0.0001 versus control group; #p < 0.05, ##p < 0.01, ###p < 0.001, ####p < 0.0001 versus oe-miR-124-3p treated group. Source data are provided as a Source data file.

dimerize via the N-terminal Qua1 domain[36]. The majority of QKI binding action occurs within introns, which include potential QKI response elements. Consistent with a role in splicing, QKI is essential for the enhanced production of many circRNAs[37] and acts by binding to recognition elements within introns, which form splice sites in the vicinity of circGlis3. Furthermore, mutation of QKI motifs fail to induce circGlis3 formation. The secondary structure within *Glis3* pre-mRNA that brings the head and the tail of exon 3 together has been demonstrated to enhance circGlis3 biogenesis. QKI probably enhances circGlis3 biogenesis by drawing the circle-forming *Glis3* exon 3 into proximity via the dimer structure, which can bind two well-separated regions of a single RNA molecule.

circGlis3 levels in islets or clinical serum samples show a trend of initially increasing and then decreasing with continuously rising glycaemia. Herein, our data showed that chronic lipotoxicity stress caused FUS-SG assembly, and that circGlis3 colocalized with FUS and was restrained in SG. SG are nonmembrane-bound assemblies formed by phase separation, and their components may include abundant mRNAs[38], nontranslating mRNAs[39], oligonucleotides[40], ribonucleoproteins[41] and translation initiation factors[42]. The structures of SG are highly dynamic, and the internal components constantly exchange with the surrounding cellular content[41]. As a type of nonmembrane-bound compartment, SG behave like condensed liquid phases of the cytoplasm or nucleoplasm, and are generally formed when cells experience stress such as oxidative stress, ER stress, and viral infection, as well as modulating the stress response[43]. According to previous studies, preventing SG formation might be a potential therapeutic strategy for treating obesity and T2DM. Saturated fatty acids, for example, serve as endogenous stressors, disrupting PDX1 nucleocytoplasmic transport by stimulating SG formation in pancreatic β-cells, and overstress results in cytoplasmic SG formation[29]. Notably, SG contain both mRNAs and ncRNAs, and further analysis of the SG transcriptome revealed that SG-enriched mRNAs differ considerably from secreted, mitochondria-localized, or ER-localized mRNAs[43]. These results indicated that when mRNAs are present in SG, they have restricted diffusion. However, little research has been done on the SG capturing circRNAs and the relationship between this dynamic behavior and disease onset.

FUS acts as an RNA-binding protein and performs critical functions in transcription, pre-mRNA splicing, RNA processing, and DNA repair. Depending on the type of reversible stress, FUS rapidly shuttles between liquid compartments in the nucleus and the cytoplasm but is predominantly localized to the nucleus in unstressed cells. The liquid-like compartments of FUS maintain the trade-off between function and the risk of cytoplasm aggregation[44]. Disease-linked FUS stimulation

can accelerate this aberrant phase transition from liquid drops to insoluble solid aggregates[45]. In SG, insoluble FUS accumulates in the cytoplasm, and simultaneous depletion of FUS from the nucleus may induce degenerative diseases[28]. In the present study, we investigated that circGlis3 was located and restricted to SG via mutual combination with FUS when cells suffered chronic lipid stress. Therefore, FUS-formed SG likely restrain circGlis3 and cause cytoplasmic free circGlis3 to diminish progressively with β-cell decompensation triggered by long-term obesity or diabetes. The circGlis3 function and downstream signal transduction mechanisms can then be suppressed.

Notably, increasing the circGlis3 level in obesity islets is beneficial for maintaining β-cell compensation by enhancing β-cell function and inhibiting β-cell apoptosis. The results demonstrated that circGlis3 increases insulin transcription and secretion in β-cells in vivo and in vitro by sponging endogenous miR-124-3p, resulting in increased *NeuroD1* and *CREB1* expression. A previous report showed that miR-124-3p in β-cells has a role in pancreatic development and glucose metabolism by regulating the downstream target gene *Foxa2*[46]. Furthermore, miR-124-3p is overexpressed in diabetic human pancreatic islets and inhibits insulin secretion[47]. Bioinformatics analysis and experiments revealed that miR-124-3p regulates multiple target genes, including *NeuroD1* and *Creb1*. *NeuroD1* is an insulin transcription factor that enhances insulin production by targeting the insulin promoter[48]. *Creb1* plays a central role in gluconeogenic regulation, lipid metabolism and insulin signaling pathways[49] and is necessary for maintenance of glucose homeostasis as well as the activation of the gluconeogenic regulatory program[50]. In addition, *Creb1* can be targeted by *miR-10a*, which regulates glucose metabolism and insulin secretion in T2DM[51]. Notably, we showed that circGlis3/miR-124-3p/*NeuroD1* and *Creb1* are involved in pathological changes in β-cells in response to obesity. Under obese conditions, miR-124-3p substantially inhibited insulin transcription and secretion of β-cells, and this inhibition was reversed when circGlis3 was overexpressed. The results amply demonstrated that circGlis3 mediates β-cell function, though they cannot absolutely deny the exist of potential extra non-beta-cell effects due to ectopic expression of circGlis3 in in vivo study. In fact, circGlis3 did not influence the fasting glucagon and somatostatin secretion, it indicated that circGlis3 exerted its effect mainly by mediating β-cell function.

Another possibility is that circGlis3 decelerates pancreatic β-cell apoptosis in mice with obesity by retarding the activation of the Caspase 3 pathway by binding to the proapoptotic factor SCOTIN. The SCOTIN gene is directly transactivated by p53, which is a transcription factor that induces growth arrest or apoptosis in response to cellular stress. The majority of SCOTIN protein is localized to the endoplasmic reticulum and can induce cell apoptosis in a Caspase

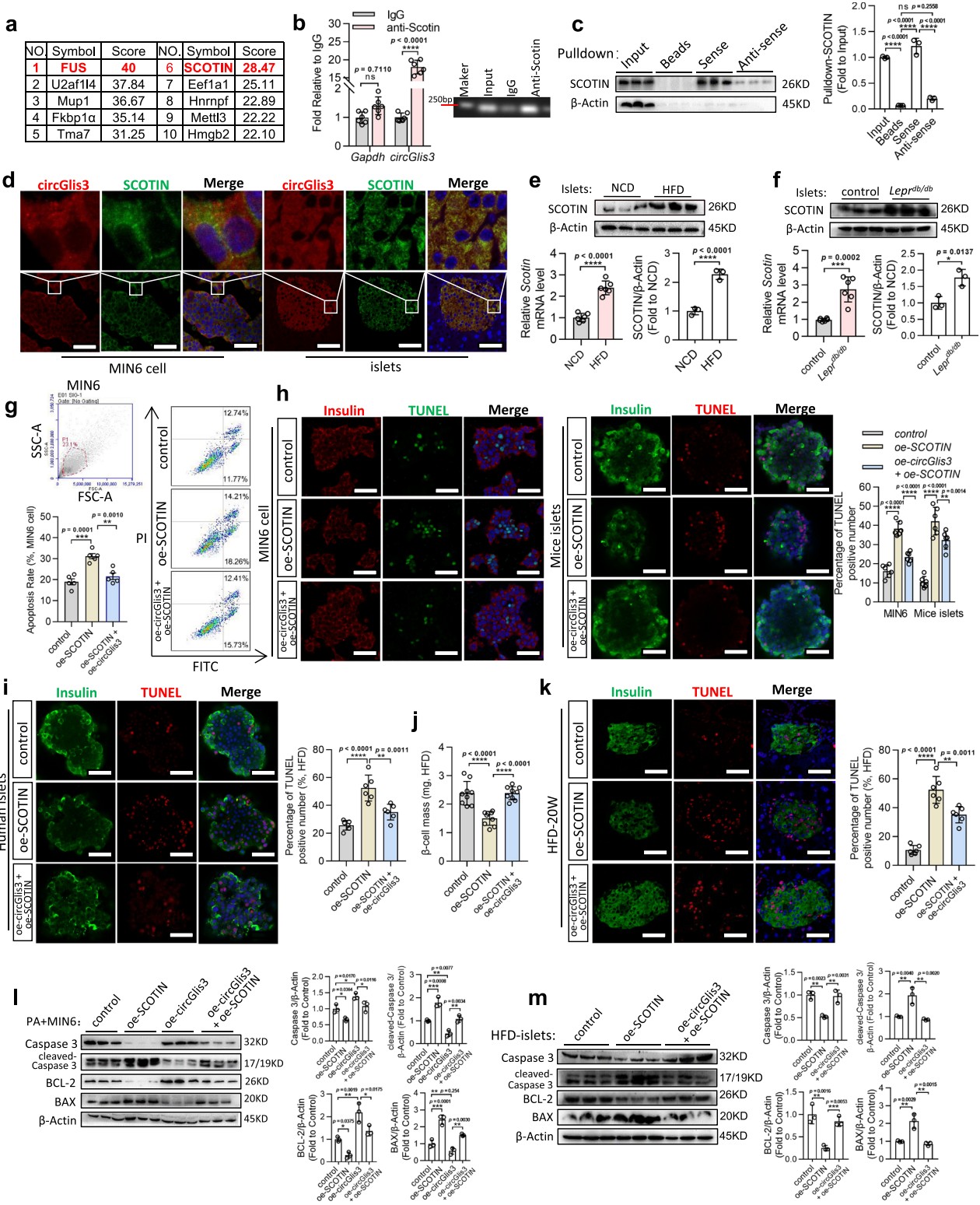

3-dependent manner[25]. β-cell apoptosis was confirmed to be a major contributor to the development of β-cell decompensation in the late stages of T2DM, with a reduction in β-cell mass of 25–50% at the time of diagnosis[52]. Furthermore, previous studies in patients with T2DM revealed that increased β-cell apoptosis resulted in decreased proliferation and β-cell mass. Since circGlis3 and SCOTIN are both located in the cytoplasm, circGlis3 may inhibit the activity of proapoptotic proteins by binding to SCOTIN and restraining the location of SCOTIN in the ER. Nevertheless, circGlis3 overexpression or knockdown had

little effect on β-cell proliferation. These results indicate that circGlis3 overexpression promotes insulin transcription and secretion by suppressing the miR-124-3p/NeuroD1 and Creb1 pathways and decreases cell apoptosis by suppressing the Caspase 3-dependant pathway through binding to and inhibiting the proapoptotic activity of SCOTIN. However, when β-cells were subjected to chronic lipid stress, free circGlis3 was captured by FUS-formed SG in the cytoplasm, inhibiting the downstream mechanism of miR-124-3p sponging and SCOTIN binding.

**Fig. 6 | circGlis3 prevents β-cell apoptosis by directly binding to SCOTIN. a** The mass spectrometry assay revealing the proteins (score > 20 by Mascot Software) pulled down by biotin-labeled circGlis3 from the MIN6 cells lysate. **b** The endogenous circGlis3 interacting with SCOTIN in MIN6 cells were detected by RIP assay and RT-PCR ($n = 6$ biological replicates). **c** Biotin-labeled sense or anti-sense circGlis3 were used for circGlis3-protein pulldown against MIN6 cells lysate, and Western Blotting showing that SCOTIN was pulled down. **d** Colocalization between circGlis3 and SCOTIN in MIN6 cell (Red represents circGlis3, Green represents SCOTIN, Scale bars represent 50 μm, $n = 5$ biological replicates) and pancreas sections (Scale bars represent 100 μm, $n = 5$ biological subjects) by circRNA fluorescent in situ hybridization (circFISH) and Immunofluorescence assays. **e, f** RT-PCR analysis and Western Blotting results of *Scotin* mRNA and protein level in the islets from the HFD-fed mice and *Lepr*[db/db] mice ($n = 6$ biological subjects). **g** Annexin V/PI staining and flow cytometry analysis of cell apoptosis in MIN6 cells with oe-circGlis3 or/and oe-SCOTIN treatment ($n = 5$ biological replicates). **h** TUNEL assays in MIN6 cell (Red represents positive TUNEL cells, Green represents Insulin; Scale bars represent 100 μm; $n = 5$ biological replicates) and mice islets (Red represents Insulin, Green represents positive TUNEL cells; Scale bars represent 50 μm; $n = 5$ biological subjects) with oe-circGlis3 or/and oe-SCOTIN treatment. **i.** TUNEL assays in human islets (Red represents positive TUNEL cells, Green represents Insulin; Scale bars represent 100 μm) with oe-circGlis3 or/and oe-SCOTIN treatment ($n = 5$ biological subjects). **j** β-cell mass in oe-circGlis3 or/and oe-*SCOTIN* treated HFD-fed mice ($n = 6$ biological animals). **k** TUNEL assays in pancreatic sections of oe-circGlis3 or/and oe-SCOTIN treated mice with HFD-fed for 20 weeks (Red represents positive TUNEL cells, Green represents Insulin; Scale bars represent 100 μm; $n = 5$ biological animals). **l, m** Western Blotting showing Caspase 3, cleaved-Caspase 3, BAX, and BCL-2 in palmitate-stimulated MIN6 cell and islets of the HFD-fed mice, both with oe-circGlis3 or/and oe-SCOTIN treatment, respectively. **b, c, e–m** For bar graphs, data represents mean ± SEM. **b, c, g–m** One-way ANOVA with Tukey's post-test. **e, f** Unpaired two-tailed Student's t-test. $^*p < 0.05$, $^{**}p < 0.01$, $^{***}p < 0.001$. Source data are provided as a Source data file.

In conclusion, our findings suggested that circGlis3 plays a crucial role in the development of obesity-associated β-cell dysfunction. Overexpression of circGlis3 upregulated insulin transcription and secretion, improved abnormal glucose tolerance and β-cell apoptosis. We proved that circGlis3 lengthened the β-cell compensated stage and delayed the T2DM process, and that circGlis3 increase will be a promising diagnostic and therapeutic target in T2DM, bringing a new dimension to the functional importance of circRNA regulation in diabetes mellitus. Certainly, there are still some limitations in our study. For example, in humans with obesity, the increase in circGlis3 expression in the serum may be partially due to a larger adipose mass, although circGlis3 levels did not result in any significant changes in the white fat of mice with obesity. And further studies are needed to determine whether serum circGlis3 levels may be used as a biomarker to detect the pathophysiological status of diabetes or obesity.

## Methods

### RNA-sequencing and analysis

The male mice used for RNA sequencing were 5 weeks old when purchased and fed HFD for 8 weeks[14]. Total islet RNA (more than 200 islets/group) was isolated using the RNeasy Kit (Qiagen). The RNA purity was checked using a Nanodrop 2000, and the integrity was assessed using an Agilent 2100 Bioanalyzer System (Agilent Technologies).

The sequencing libraries were prepared following the manufacturer's recommendations of the VAHTSTM Total RNA-seq (H/M/R) Library Prep Kit for Illumina. After cluster generation, the libraries were sequenced on an Illumina Hiseq X10 platform, and 150-bp paired-end reads were generated. The raw reads were filtered by removing reads containing adapter, poly-N and low-quality reads for subsequent analysis.

The clean reads were aligned to the reference genome by Bowtie2[53]. Then for unmapped reads, the junctions were picked out using back-splice algorithm and circRNAs were verified with circRNA Finder[54] (https://github.com/bioxfu/circRNAFinder). Finally, circRNAs were annotated and abstracted with circRNA Anno of circRNAFinder which were considered as the reference sequence for further analysis. Differentially expressed circRNA were analyzed using DESeq2 package based on the negative binomial distribution test[55]. The thresholds of differentially expressed cirRNAs: the absolute value of log2 (fold change) ≥ 1 and false discovery rate analysis for multiple testing (FDR) ≤ 0.05.

### Serum samples and human islets

Human islets were provided by the Tianjin First Central Hospital. Pure human islets (>80%) were collected and cultured in CMRL-1066 medium (Corning) with 10% human serum albumin (Baxter), 100 U/mL penicillin, and 100 μg/mL streptomycin and incubated at 37 °C under a 5% $CO_2$ atmosphere.

The clinical serum and clinicopathological data were collected from Zhongda Hospital, Affiliated with Southeast University (Nanjing, China). All the patients enrolled in this study were considered obese (BMI > 24). The negative controls were individuals without obesity (BMI ≤ 24). 72 male and 46 female were absorbed in this study ultimately.

All human subjects provided informed consent. All human studies were conducted in accordance with the principles of the Declaration of Helsinki and approved by the Ethics Committees of the Department Zhongda Hospital Southeast University (Nanjing, China, 2018ZDSYLL132-P01).

### Animal experiments

Since female mice must be tested across the estrous cycle and are more variable than males, only male mice were used in this study. The male C57BL/6 (age: 7–8 weeks) and *Lepr*[db/db] (age: 4–5 weeks) mice used for animal experiments were purchased from the Model Animal Research Center of Nanjing University. All mice were housed 3–5 animals per cage, maintaining on a 12 h light and dark cycle with free access to water, at room temperature (25 °C) and in the indoor humidity 50–60%. The C57BL/6 J mice were fed either a normal diet consisting of standard laboratory chow or a high fat diet at 8 weeks of age (D12494, 60% energy from fat) for at least 3 months to establish a T2DM model. All animal experiments were performed in strict accordance with the guidelines and rules formulated by the animal ethics committee of China Pharmaceutical University (Nanjing, China; Permit No. 2162326).

After all experiments, the mice were fasted overnight and then euthanized via intraperitoneal injection of sodium pentobarbital overdose. Blood sample was collected and serum was isolated. Pancreas tissues and pure islets were collected separately after euthanasia.

### Construction of the circGlis3 overexpression vector

circGlis3 was amplified from mouse genomic DNA, and the sequence of exon 3 of the *Glis3* gene was inserted into a pEx-3 ciR vector (GenePharma Co., Shanghai, China) and a pAV-insulin1-circRNA-EFFS-GFP vector (ViGene Biosciences Co., Shandong, China), which both contained circular structures. All of the constructs were confirmed by sequencing. The circGlis3 sequence is listed in Supplementary Table 1.

### Pancreatic intraductal viral infusion

Male C57BL/6 and *Lepr*[db/db] mice were injected with adeno-associated virus serotype 8 vector (AAV8)-MIP (mouse insulin1 promoter)-circGlis3 (ViGene Biosciences), lentiviral vector up-LV2N-mmu-miR-124-3p-Puro (Gene Pharma), lentiviral vector up-LV-MIP-Fus-Puro (Corues Biotechnology) or control virus. The viruses ($10^{12}$ GCP/mL) were infused at a rate of 6 μL/min via pancreatic intraductal viral infusion as detailed elsewhere[56].

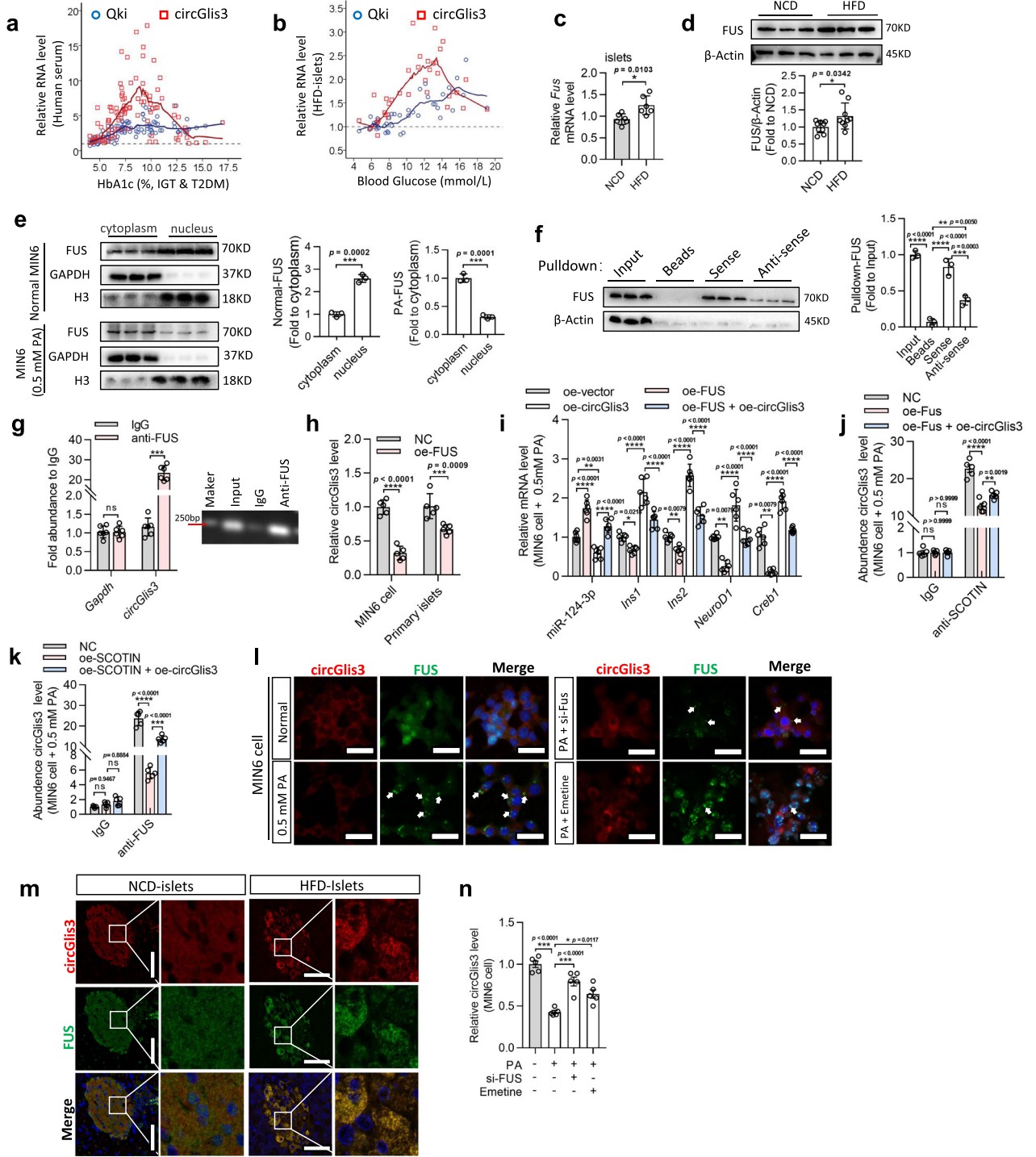

## Glucose and insulin tolerance tests in vivo

After viral infusion for 2 weeks, glucose and insulin tolerance tests were conducted in the experimental mice. After measurement of the fasting blood glucose level, IPGTTs were performed by using an injection of 2 g/kg body weight glucose (Sigma), and the blood glucose level was determined after 5, 15, 30, 60, 90, and 120 min. For the IPITT, the mice were intraperitoneally injected with 1 U/kg bodyweight insulin after 12 h of overnight fasting. The serum sample was separated from the orbital venous blood at 0, 5, 15, and 30 min. The insulin level was detected by using a Mouse Insulin ELISA Kit (Crystal Chem) according to the manufacturer's instructions.

## Isolation, culture and dissociation of mouse islets

After euthanasia, the mouse islets were isolated by using the intra-ductal collagenase technique[8,57] and the digestion process was terminated with KRBH balanced buffer (composition: 115 mM NaCl, 4.8 mM KCl, 2.5 mM $CaCl_2$, 1.2 mM $MgSO_4$, 1.2 mM $KH_2PO_4$, 20 mM $NaHCO_3$, and 16 mM HEPES; pH 7.4) with 0.1% bovine serum albumin, 2.5 mM glucose and 1% penicillin/streptomycin. Then, the islets were hand-picked and incubated overnight in 1640 medium containing 10% FBS and 1% penicillin/streptomycin. After incubation overnight in 1640 medium, the islets were seeded in the plates used for subsequent experiments.

**Fig. 7 | FUS sequestrates circGlis3 to reduce its abundance in diabetes by assembling cytoplasmic SG. a** Expression of circGlis3 and *Qki* mRNA in patients with IGT or type 2 diabetes mellitus (T2DM) ($n$ = 98 human individuals). **b** Expression of circGlis3 and *Qki* mRNA in HFD-fed mice with obesity or T2DM ($n$ = 40 biological animals). **c** RT-PCR analysis of FUS mRNA expression in islets of the NCD- and HFD-fed mice ($n$ = 6 biological subjects). **d** Western Blotting showing FUS protein expression in islets of the NCD- and HFD-fed mice. **e** Western Blotting showing FUS shuttle between the nucleus and cytoplasm in responses to palmitate stress in MIN6 cells. **f** Biotin-labeled sense or anti-sense circGlis3 were used for RNA-protein pulldown against MIN6 cells lysate, and Western Blotting showing that FUS was pulled down. **g** The endogenous circGlis3 interacting with FUS in MIN6 cells were detected by RIP assay and RT-PCR ($n$ = 6 biological replicates). **h** RT-PCR analysis of circGlis3 expression in MIN6 cells ($n$ = 6 biological replicates) and islets with FUS overexpression ($n$ = 5 biological subjects). **i** RT-PCR analysis of miR-124-3p, *Ins1, Ins2, NeuroD1* and *Creb1* mRNA in MIN6 cells with FUS and circGlis3 overexpression ($n$ = 6 biological replicates). **j** RIP and RT-PCR assay analyzed fold

enrichment of endogenous circGlis3 by SCOTIN in palmitate-stimulated and oe-FUS-treated MIN6 cells ($n$ = 6 biological replicates). **k** RIP and RT-PCR assay analyzed fold enrichment of endogenous circGlis3 by FUS in palmitate-stimulated and oe-SCOTIN-treated MIN6 cells ($n$ = 6 biological replicates). **l** Colocalization between circGlis3 and FUS-stress granules (FUS-SG) in the palmitate-stimulated MIN6 cells with *si-Fus* and Emetine treatment (Red represents circGlis3, Green represents FUS; Scale bars represent 20 μm, $n$ = 5 biological replicates). **m.** Colocalization between circGlis3 and FUS-SG in islets of the HFD-fed mice (Red represents circGlis3, Green represents FUS; Scale bars represent 100 μm, $n$ = 6 biological animals). **n.** RT-PCR analysis of circGlis3 level in the palmitate-stimulated MIN6 cells with si-FUS and Emetine treatment ($n$ = 6 biological replicates). **c–k, n** For bar graphs, data represents mean ± SEM. **a, b** LOESS Curve Fitting, 35% of points to fit. **c–e** Unpaired two-tailed Student's *t*-test. **f, n** One-way ANOVA with Tukey's post-test. **g–k** Two-way ANOVA with Bonferroni's post-test. $^*p < 0.05$, $^{**}p < 0.01$, $^{***}p < 0.001$, $^{****}p < 0.0001$. Source data are provided as a Source data file.

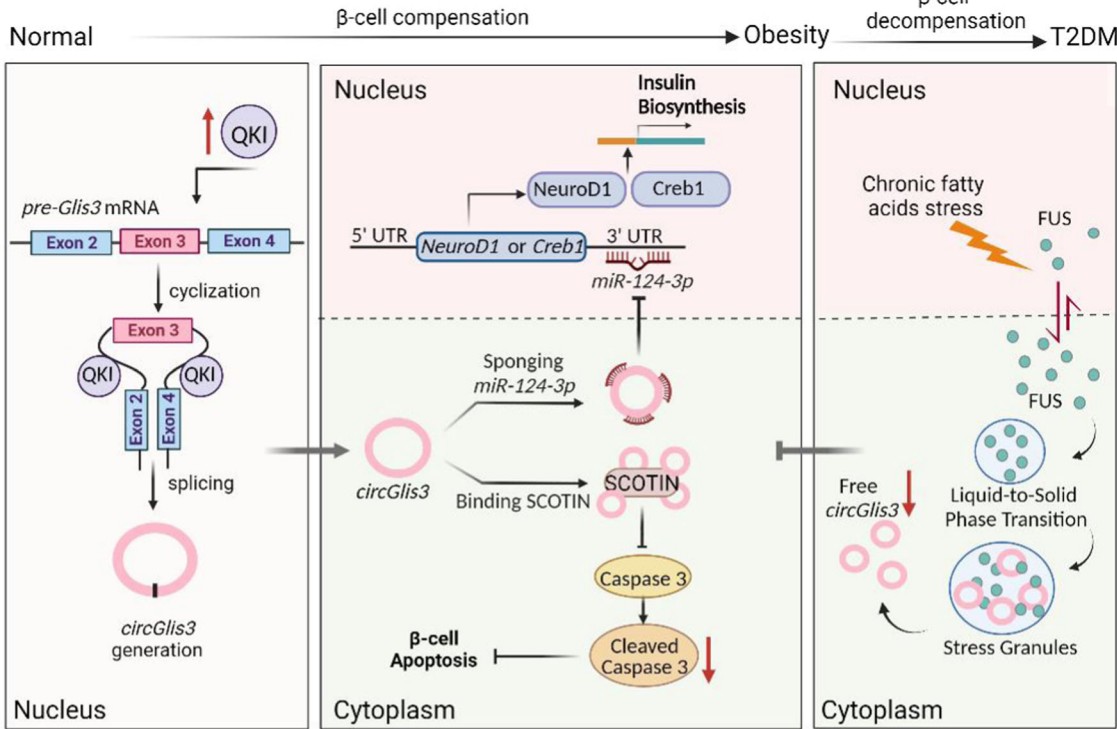

**Fig. 8 | A working model that illustrates the mechanism by which circGlis3 regulates β-cell function.** circGlis3 is regulated by QKI-mediated splicing and increases in individuals with obesity and moderately diabetes, while it decreases when diabetes occurs via FUS-formed SG. circGlis3 functions in β-cell compensation through regulating insulin transcription and apoptosis by sponging miR-124-3p and directly binding SCOTIN provide a potential therapeutic angle.

### Insulin secretion and content assay

A total of 30 islets/well were seeded in a 96-well plate, and MIN6 cells were seeded in a 48-well plate. Then, the cells were transfected with circGlis3 overexpression plasmid or circGlis3 knockdown plasmid using Lipo2000 in accordance with the manufacturer's instructions and cultured for 48 h.

For GSIS, islets or MIN6 cells were incubated with glucose (2.5 mM) for 4 h prior to incubation with glucose (2.5 mM or 16.7 mM) for 2 h.

For palmitate-stimulated insulin secretion, islets or MIN6 cells were incubated with palmitate (0.5 mM) and glucose (2.5 mM) for 4 h prior to incubation with glucose (2.5 mM or 16.7 mM) for 2 h.

The supernatants were collected and measured by using the Mouse Insulin ELISA Kit (Mercodia) according to the manufacturer's instructions. The insulin secretion level was normalized to the total cellular protein content. For insulin content, the islets and MIN6 cells were extracted with acid ethanol. The insulin concentration of the

extraction was measured by using the Mouse Insulin ELISA Kit and normalized by the total cellular protein content. The protein contents of mouse islets or MIN6 cells were quantified by using a BCA kit (Vazyme).

### Glucagon and somatostatin secretion in islet cells

A total of 30 islets/well were seeded in a 96-well plate and transfected with either the circGlis3 overexpression plasmid or circGlis3 knockdown plasmid and cultured for 48 h. Then, the islets were pre-incubated for 30 min at 37 °C in KRBH balanced buffer and supplemented with 0.1% bovine serum albumin and 2.5 mM glucose. Subsequently, the buffer was changed, and the islets were incubated with 2.5 mM or 16.7 mM glucose for 2 h. Immediately after incubation, aliquots of the medium were removed for glucagon and somatostatin assays. The islets were then lysed via ultrasonication to extract the total cellular glucagon content. Secreted glucagon and somatostatin in the supernatants or lysates were quantified using either a

mouse glucagon ELISA kit or somatostatin ELISA kit (both from Mercodia).

## HOMA-IR

The homeostasis model assessment method was applied to calculate insulin resistance (HOMA-IR), as described in Johnson's study[58]. The fasting plasma glucose and insulin concentrations were calculated as follows: fasting plasma glucose (mmol/L) × fasting plasma insulin (mU/L)/22.5.

## Total RNA isolation and real-time reverse transcription-PCR (RT-PCR)

Total RNA was first extracted from the cell lines or tissues by using TRIzol (Solarbio) based on the guidebook. For mRNA or circRNA amplification, the RNA was reverse transcribed into cDNA by using a reverse transcriptase kit (ABM). Real-time PCR was performed by using the LC480 Light Cycler (Roche) with qPCR SYBR Green Master Mix (Vazyme). For circRNA amplification, divergent primer sets were applied.

For miRNA amplification, reverse transcription and real-time PCR were performed using the Hairpin-itTM miRNAs qPCR Quantitation Kit (Gene Pharma) according to the manufacturer's instructions.

All miRNA and mRNA levels were normalized to the expression of small RNAs (*sno234* and *U6*) or mRNA (*Gapdh* and *β-Actin*), respectively. All the primers used in this study are listed in Supplementary Table 1.

## RNase R digestion

Total RNA was either treated or not (control) with RNase R (4 U/ug) (Epicenter) in the presence of 1 × Reaction Buffer and incubated for 20 min at 37 °C. Then, reverse transcription and RT-PCR were performed as described in the RNA extraction and RT-PCR section.

## Culture of MIN6 cells

The murine insulin-secreting cell line MIN6 was cultured in DMEM-Glutamax medium (Invitrogen) with 15% foetal calf serum (Gibco), 1% penicillin/streptomycin (Gibco), and 50 μM β-mercaptoethanol (Sigma). The MIN6 cells tested negative for mycoplasma contamination. The culture plates were maintained in a humidified incubator with 5% $CO_2$ at 37 °C. For palmitate treatment, the islets and MIN6 cells were incubated in 0.5 mM palmitate (Sigma). For glucose treatment, the islets and MIN6 cells were incubated in 33.3 mM glucose.

## Cell transfection

Transfection of the dissociated mouse islets or MIN6 cells was performed using Lipofectamine 2000 (Invitrogen) for plasmids, shRNA, siRNA sequences and miRNA mimics. The plasmids, shRNA, siRNA sequences, and miRNA mimics used in this study are listed in Supplemental Table 1. All assays were performed 48 h after transfection.

## Isolation of cytoplasmic and nuclear fractions

MIN6 cells (at least $10^8$) were harvested and treated using the PARIS KIT protein and RNA isolation system (Invitrogen) according to the manufacturer's instructions. *β-Actin* and *U6* served as controls for cytoplasmic RNA and nuclear RNA, respectively.

## β-cell and α-cell labeling and sorting and measurement of the level of circGlis3

The mouse islets were isolated using the intraductal collagenase technique. The pure islets (more than 600 islets isolated from 5-6 mice that were used as biological replicates) were digested into single cells with 0.125% trypsin solution (without EDTA). After washing in ice-cold PBS by centrifugation at 4 °C and 500 x *g* for 10 min, the single cells were fixed in fixative solution (flow cytometry, Shanghai Huzheng) at room temperature for 10 min. After two washes with PBS, the cells were permeabilized with permeabilization wash buffer (Yeasen) for 5 min. After preincubation with the blocking solution (20% mouse serum) at 4 °C overnight, mouse anti-insulin (1:100) and rabbit anti-glucagon (1:100) were both added to the cells and incubated at 37 °C for 1 h. After two washes with ice-cold PBS, goat anti-mouse Alexa Fluor 488 (1:500) and donkey anti-rabbit Alexa Fluor 647 (1:500) were both added to the cells and incubated at 37 °C for 1 h.

The cells were washed twice with ice-cold PBS and sorted by using BD FACSAria II SORP, and gating was performed using BD FACSDiva software (Becton, Dickinson Biosciences). Cell sorting was performed with a 100 μm nozzle size and sorted directly into 5 mL tubes containing 3 mL of PBS to minimize cellular stress. Cells (10,000-100,000) of each population of interest were sorted at a speed of 1500 cells/s.

After the cells were washed in ice-cold PBS by centrifugation at 4 °C and 500 x *g* for 10 min, they were resuspended in 1 mL of TRIzol and prepared for RNA extraction. The level of circGlis3 in sorted β-cells and α-cells was detected by RT-PCR assays.

## CCK-8 analysis

The proliferation of MIN6 cells was measured by the CCK-8 assay (Vazyme). Then, $5 \times 10^3$ cells/100 μL were incubated in 96-well plates and cultured in conditioned medium. Each sample was assayed in triplicate. At 0, 12, 24, 48, and 72 h, 10 μL of the CCK-8 reagent was added to each well and the plate was incubated at 37 °C for 1 h. After incubation, the absorbance at 450 nm was measured using an automatic microplate reader (BioTek).

## β-cell mass

The mice were fasted overnight and pancreas samples were collected immediately after euthanasia. The β-cell mass for each mouse was measured by first obtaining the fraction of the cross-sectional area of the pancreatic tissues (i.e., exocrine and endocrine) positive for insulin staining and then multiplying this by the pancreatic weight[59].

## Apoptosis analysis

For flow cytometric analysis, MIN6 cells were transfected and collected after 48 h. Cell apoptosis was determined by using the Annexin V/FITC Apoptosis Detection Kit (Beyotime) according to the manufacturer's instructions. FITC and PI were detected at 488 nm and 630 nm, respectively. Data were analyzed by BD Accuri C6 Software and FlowJo v10 software.

For TUNEL staining, the DNA fragmentation of islets or MIN6 cells was determined by using a TUNEL assay kit (Beyotime). Islets or MIN6 cells were fixed in 4% paraformaldehyde at room temperature for 15 min, washed with PBS three times, and permeabilized with 0.3% Triton-X for 15 min. After preincubation with the blocking solution (1% BSA) for 1 h, rabbit anti-insulin (1:200) was incubated at 4 °C overnight, and goat anti-rabbit Alexa Fluor 647 (1:500) was incubated at 37 °C for 1 h. The islets or MIN6 cells were treated with TUNEL staining and observed under an LSM510 Zeiss confocal microscope (Zeiss).

## Western blots

The whole cell lysate and nuclear-protein fractions were isolated from the tissues or cells using a protein extraction kit (Beyotime). Pancreatic tissues, primary islets (200 islets per group), and MIN6 cells were ruptured with RIPA buffer (Beyotime) containing 1% PMSF (Sigma). The protein concentration was quantified by using a BCA kit (VAzyme). The protein samples were resolved by SDS–PAGE and transferred onto PVDF membranes, blocked for 2 h with 5% skim milk, and incubated overnight at 4 °C with primary antibodies. After three washes with TBST, the membranes were incubated for 1 h with the appropriate secondary antibodies and developed using chemiluminescent substrates. All the antibodies used in this study are listed in Supplementary Table 2. Images of blots were collected by Tanon 3500 system, the gray value was calculated by ImageJ (v1.8.0).

## RNA FISH

For determination of the subcellular location of circRNA and miRNA, hybridization was performed using a fluorescent in situ hybridization kit purchased from Gene Pharma. Cy3-labeled circGlis3 and FAM-labeled mmu-miR-124-3p were incubated. The samples were analyzed on an LSM510 Zeiss confocal microscope (Zeiss). The Cy3-labeled circGlis3 used for RNA in situ hybridization was 5′-GCAAAA + TAAC GGACCGTACACT + TGT-3′, and the FAM-labeled miR-124-3p used for RNA in situ hybridization was 5′-GGCAT + TCACCGCGTGCCT + TA -3′.

## Immunofluorescence

MIN6 cells were fixed with 4% paraformaldehyde, pancreatic sections were deparaffinized, and tissue antigens were retrieved. The cells and pancreatic sections were permeabilized with 0.3% Triton X-100 for 15 min and then blocked with 1% BSA at 37 °C for 1 h. After incubation with the primary antibodies at 4 °C overnight, Alexa Fluor-conjugated secondary antibodies (Abcam) were added, and the plate was incubated at 37 °C for 1 h. Next, the cell nuclei were stained with DAPI (Beyotime) for 10 min at room temperature. Images were obtained by Carl Zeiss LSM700 and analysed by ZEN 2012. All the antibodies used in this study are listed in Supplementary Table 2.

## RNA-binding protein immunoprecipitation assay

RNA immunoprecipitation (RIP) experiments were performed by using the Magna RIP RNA-Binding Protein Immunoprecipitation Kit (Millipore) according to the manufacturer's instructions. Approximately $1 \times 10^8$ cells were pelleted and resuspended with an equal pellet volume of RIP lysis buffer (100 μL) plus protease and RNase inhibitors. The cell lysates were then incubated with 5 mg of the control mouse IgG or antibody-coated beads with rotation at 4 °C overnight. After treatment with proteinase K, the immunoprecipitated RNAs were extracted through phenol–chloroform extraction.

## RNA pulldown assay and mass spectrometry analysis

RNA pulldown assays were performed using the Magnetic RNA-Protein Pull-Down Kit (Thermo). Briefly, the cell lysates were prepared by ultrasonication in RIP buffer (150 mM KCl, 25 mM Tris [pH 7.4], 0.5 mM dithiothreitol, 0.5% NP-40, complete protease inhibitor cocktail, and RNase inhibitors) and precleared against streptavidin magnetic beads (Invitrogen). In vitro transcribed biotin-labeled RNA adsorbed to streptavidin in the magnetic beads were incubated with cell lysates at 4 °C for 4 h before five washes with RIP buffer and elution in Laemmli sample buffer. Mass spectrometric analysis of the RNA-binding protein samples was entrusted to Shanghai Applied Protein Technology Co., Ltd. (Shanghai, China).

## In vitro transcription

The DNA template used for the in vitro synthesis of biotinylated circGlis3 was generated by PCR. The forward primer contained the T7 RNA polymerase promoter sequence to allow for subsequent in vitro transcription. The PCR products were purified using the DNA Gel Extraction Kit (Vazyme), and in vitro transcription was performed using the Transcript Aid T7 High Yield Transcription Kit (Thermo) and RNA 3′ End Biotinylation Kit (Thermo) according to the manufacturer's instructions. RNA was subsequently purified through phenol–chloroform extraction. The primer sequences used in this study are listed in Supplementary Table 1.

## In vitro cyclization

In vitro cyclization of linear RNA was performed as suggested by Qidong Li, albeit with some modifications[60]. For assembly of the pre-ligation complexes, both biotin-labeled and unlabeled linear RNA were incubated with the indicated DNA splints (at a molar ratio of 1:1.5) at 90 °C for 2 min, followed by cooling to room temperature over 15 min. Ligation to form circular RNA was performed by incubation with T4 DNA ligase (TaKaRa) at 16 °C overnight. Next, the RNA was treated with RNase R and DNase I at 37 °C for 30 min to degrade the residual linear RNA and DNA, respectively. RNA was finally purified through phenol–chloroform extraction. The primers and DNA splint sequences used in this study are listed in Supplementary Table 1.

## Mutation of QREs

We conducted this experiment based on a past report[20], albeit with some modifications. Briefly, the genomic regions comprising Glis3 exon 3 two introns were synthesized (Tsingke Biotechnology) with WT QRE sequences or mutated QREs (1, 2, 3, and 4, individually and synthetically) approximately 500 bp away from the splicing site of the central exon. These regions were then cloned into pcDNA3.1 and transfected into MIN6 cells that had been transfected 24 h previously with the control or QKI overexpression plasmid. Then, RNA was isolated 24 h later with TRIzol and treated with RNase R (Epicentre). cDNA was reverse transcribed, and RT-PCR was performed to analyze the ratio of circGlis3 to *Glis3*.

## Luciferase assay

The pMIR-REPORT vector (Promega) with full-length circGlis3 or the 3′-UTR of *Creb1* and *NeuroD1* was used for the luciferase reporter assays. The binding sites of miR-124-3p in circGlis3 or the 3′-UTR of *Creb1* and *NeuroD1* were mutated for mutant vectors. MIN6 cells were plated at a concentration of 10,000 cells per well in a 24-well plate and cultured overnight. The cells were then transfected with 0.6 μg DNA per well (0.5 μg construct promoter and 0.1 μg constitutive Renilla expression plasmid as a control for transfection efficiency) and miR-124-3p mimics or inhibitors. These cells were then harvested 24 h after transfection, and the luciferase activities were measured by using a dual-luciferase reporter assay system (Promega).

## Quantification and statistical analysis

For all in vivo, in vitro, and human sample studies, $n$ represents the number of biological replicates per group (as detailed in the Figure Legends) in accordance with the actual situation. No statistical methods were applied to predetermine the sample size. The data are presented as the means ± S.E.M.s. Statistical analysis was performed using SPSS 22.0 and GraphPad Prism 7 to assess the differences between the experimental groups. Statistical significance was determined by two-tailed Student's $t$ test for two groups, one-way ANOVA with Tukey's post-test for univariate comparisons, and two-way ANOVA with Bonferroni's post-test for bivariate comparisons. Correlation analysis was evaluated using Pearson correlation and regression analysis. The statistical significance was set at $^*p < 0.05$, $^{**}p < 0.01$, and $^{***}p < 0.001$.

## Reporting summary

Further information on research design is available in the Nature Portfolio Reporting Summary linked to this article.

## Data availability

CircGlis3 was recorded in circBase database (http://circbase.org/cgi-bin/simplesearch.cgi). The RNA-seq data that support the findings of this study are available at Gene Expression Omnibus (GEO), with the accession number GSE139991, Sequence Read Archive (SRA) database, with accession number PRJNA835620. The source data underlying Fig. 1a-i; 2d-e, g-k, m-r; 3a-m; 4b-s; 5b-e, g-v; 6b-c, e-m; 7a-k, n. Supplementary figs. 1a-p; 2b-h; 3a-i; 4a-j; 5a-b, d-f, k-n; 6c-j; 7b-i was provided as source data file. The RNA sequencing data of the HFD-fed (GSE139991) has been previously published[14]. All the rest of the data is newly generated. Source data are provided with this paper.

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

## Acknowledgements

This work was supported by the National Natural Science Foundation of China: Grant No. 82070801 (To L.J.), 82100858 (To FF.Z.), 82073227 (To Y.P). Supported by Natural Science Foundation of Jiangsu Province, BK20221520 (To L.J.), BK20200569 (To FF.Z.). Supported by grants from the '111' project, B16046 (To L.J.). Supported by the Priority Academic Program Development of Jiangsu Higher Education Institutions, PAPD (To L.J.). Supported by China Postdoctoral Science Foundation, 2022T150726 & 2020M671661 (To FF.Z.). Supported by Jiangsu Province Research Founding for Postdoctoral, 1412000016 (To FF.Z.). We thank the research platform (Public Platform of State Key Laboratory of Natural Medicines, China Pharmaceutical University) for the assistance with the use of the LSM700 laser confocal microscope and immunofluorescence image acquisition. We thank Yumeng Shen (Public Platform of State Key Laboratory of Natural Medicines, China Pharmaceutical University) for her assistance with flow analysis and single-cell sorting.

## Author contributions

Y.L., Y.Y., CY.X., and B.H. performed the experiments; JX.L. and JL.C. performed partial experiments on animals; GQ.L. and Q.W. collected all the human samples; Y.P. and YF.Z. analyzed data; L.L., FF.Z., and L.J. designed the project, Y.L., S.J.P., and L.J. interpreted the data and wrote the manuscript.

## Competing interests

The authors declare no competing interests.
