## [Peer Review File · Nature Communications]

Circular RNA circGlis3 protects against islet β -cell dysfunction and apoptosis in obesityEditorial Note: This manuscript has been previously reviewed at another journal that is not operating a transparent peer review scheme. This document only contains reviewer comments and rebuttal letters for versions considered at *Nature Communications*. Mentions of prior referee reports have been redacted.

REVIEWERS' COMMENTS

Reviewer #1 (Remarks to the Author):

The authors have conveniently answered my remaining criticisms.

Reviewer #2 (Remarks to the Author):

I have no further comments.

Reviewer #3 (Remarks to the Author):

Manuscript background information:

[REDACTED].

Remarks to the author:

The authors have now provided new in vitro and in vivo data to strengthen their conclusions. However, it remains uncertain whether circGlis3 effects on glucose excursion are solely mediated by beta cells. The effects presented in vitro (Fig. 3F) are rather modest and may not completely account for the glucose clearance data (Fig. 4F). The slightly enhance insulin sensitivity (Fig. 4G) and secretion (Fig. 4F) may contribute but authors might want to highlight in the discussion potential extra non-beta-cell effects due to ectopic expression of circGlis3 in gain-of-function studies.

Dear reviewers:

Thanks very much for your patience and giving us a second chance to our manuscript entitled **“Circular RNA *circGlis3* protects against islet β -cell dysfunction and apoptosis during obesity” (NCOMMS-22-35651-T).**

We accept the reviewers' new comments and agree to revise our manuscript. The manuscript has been subjected to revision carefully and accordingly. We present the comments of each reviewer below. The comments are shown in *italics*, our responses are shown in **blue font**. A thorough, point-by-point response to each point was raised and all changes, a word file of the revised manuscript with all changes has been uploaded. If you have any further questions about the revision, please do not hesitate to contact me.

Best regards,

Liang jin

Reviewers' comments:

Reviewer #1 (Remarks to the Author):

The authors have conveniently answered my remaining criticisms.

Response: Thank you for your kindness.

Reviewer #2 (Remarks to the Author):

I have no further comments.

Response: Thank you for your kindness.

Reviewer #3 (Remarks to the Author):

Manuscript background information: [REDACTED]

Remarks to the author:

*The authors have now provided new *in vitro* and *in vivo* data to strengthen their conclusions. However, it remains uncertain whether *circGlis3* effects on glucose excursion are solely mediated by beta cells. The effects presented *in vitro* (Fig. 3F) are rather modest and may not completely account for the glucose clearance data (Fig. 4F). The slightly enhance insulin sensitivity (Fig. 4G) and secretion (Fig. 4F) may contribute but authors might want to highlight in the discussion potential extra non-beta-cell effects due to ectopic expression of *circGlis3* in gain-of-function studies.*

Response: Thank you for your comments.

Insulin secretion in *oe-circGlis3*-treated primary mice islets and MIN6 cells were 1.22-fold (**Fig. 3f**) and 1.31-fold (**Fig. 3g**) higher than those in control group, respectively. Compared with negative control group, insulin secretion in *sh-circGlis3*-treated primary mice islets and MIN6 cells decreased to 81% (**Fig. 3f**) and 76% (**Fig. 3g**) respectively. It is indeed statistically significant, although the effects presented *in vitro* were seemingly modest.

In *in vivo* experiments, the adeno-associated virus serotype 8 vector (AAV8)-MIP (mouse insulin1 promoter)-*circGlis3* was injected into pancreatic duct directly. The adenovirus vectors carried mouse *insulin1* promoter, and the results of β -cell and α -cell sorting and measurement of the level of *circGlis3* also confirmed that *circGlis3* was especially enriched in β -cell (**Supplementary Fig. 1m**). Besides what's mentioned above, *oe-circGlis3* treatment did not influence the fasting glucagon (**Supplementary Fig. 4g**) and somatostatin secretion (**Supplementary Fig. 4h**) in HFD-fed mice. It means that *oe-circGlis3* treatment did not affect α cells or δ cells function.

However, *in vivo* data revealed that insulin secretion in the *oe-circGlis3*-treated mice after 8 and 20 weeks of HFD feeding were 1.50-fold (HFD feeding for 8 weeks, **Fig. 4i**) and 1.54-fold (HFD feeding for 16 weeks, **Fig. 4j**) higher than those in control animals, respectively.

Based on the above analysis, it is believed that *circGlis3* effects on glucose excursion through mediating β -cells function. Though many testing results were carried out to support our idea, we cannot absolutely deny the exist of potential extra non-beta-cell effects due to ectopic expression of *circGlis3* in gain-of-function studies. According to your advice, we added it to the discussion.

Fig. 3 e-f. Insulin secretion in glucose (2.5 mM and 16.7 mM) stimulated MIN6 cells ($n = 6$ biological replicates) and mouse islets ($n = 5$ biological subjects).

Fig. 4 i-j. Insulin secretion in glucose (2.5 mM and 16.7 mM) stimulated islets from mice with HFD-fed for 8 weeks and 16 weeks ($n = 6$ biological subjects).

Supplementary Fig. 3

Supplementary Fig. 4 g. The glucagon levels in fasting serum of HFD-fed mice ($n = 6$ biological subjects). h. Glucose-stimulated (16.7 mM) somatostatin secretion in islets from HFD-fed mice ($n = 6$ biological subjects).

The main description in the revised manuscript is as follows:

.....The results amply demonstrated that *circGlis3* mediates β -cell function, though they cannot absolutely deny the exist of potential extra non-beta-cell effects due to ectopic expression of *circGlis3* in *in vivo* study. In fact *circGlis3* did not influence the fasting glucagon and somatostatin secretion, it indicated that *circGlis3* exert its effect mainly by mediating β -cell function.